# Intracavity optical trapping of microscopic particles in a ring-cavity fiber laser

Fatemeh Kalantarifard[1], Parviz Elahi [1,2], Ghaith Makey [1,2], Onofrio M. Maragò [3], F. Ömer Ilday [1,2,4] & Giovanni Volpe [1,2,5]

Standard optical tweezers rely on optical forces arising when a focused laser beam interacts with a microscopic particle: scattering forces, pushing the particle along the beam direction, and gradient forces, attracting it towards the high-intensity focal spot. Importantly, the incoming laser beam is not affected by the particle position because the particle is outside the laser cavity. Here, we demonstrate that intracavity nonlinear feedback forces emerge when the particle is placed inside the optical cavity, resulting in orders-of-magnitude higher confinement along the three axes per unit laser intensity on the sample. This scheme allows trapping at very low numerical apertures and reduces the laser intensity to which the particle is exposed by two orders of magnitude compared to a standard 3D optical tweezers. These results are highly relevant for many applications requiring manipulation of samples that are subject to photodamage, such as in biophysics and nanosciences.

[1] Department of Physics, Bilkent University, Ankara 06800, Turkey. [2] UNAM – National Nanotechnology Research Center and Institute of Material Science and Nanotechnology, Bilkent University, Ankara 06800, Turkey. [3] CNR-IPCF, Istituto per i Processi Chimico-Fisici, 98158 Messina, Italy. [4] Department of Electrical and Electronics Engineering, Bilkent University, Ankara 06800, Turkey. [5] Department of Physics, University of Gothenburg, 41296 Gothenburg, Sweden. Correspondence and requests for materials should be addressed to F.Ö.I. (email: ilday@bilkent.edu.tr) or G.V. (email: giovanni.volpe@physics.gu.se)

**O**ptical tweezers are a powerful technique to manipulate microscopic particles[1–3] and have found applications in various research fields, from biology[4] and spectroscopy[5] to statistical physics[6] and nanosciences[7]. Standard optical tweezers consist of a single, typically Gaussian, beam focused by a microscope objective with a high numerical aperture (NA)[3,8]. A microscopic particle whose refractive index is higher than that of the embedding medium can be trapped near the focal spot because of the emergence of scattering and gradient optical forces[3]. The scattering forces are due to the radiation pressure of the light beam and act in the direction of propagation of the beam. The gradient forces push the particle toward the high-intensity focal spot. To provide gradient forces strong enough to stably trap a particle, typically water- or oil-immersion objectives with NA > 1.20 are used[3,8]. Importantly, in standard optical tweezers, the laser emission is independent of the position of the particle, which is outside the laser cavity; this corresponds to an open-loop control.

The possibility of implementing an intrinsic closed-loop control, where intracavity nonlinear feedback forces emerge when the microparticle is placed within the laser cavity, has not been exploited until now, even though we put forward this idea some years ago in some conference proceedings[9,10]. Feedback mechanisms are pervasive in science and technology, in particular linear feedback is used in resonant radiation pressure waveguides[11,12], haptic optical tweezers[13], cavity optomechanics[14], laser cooling of single atoms[15], and recently in near-field trapping in liquid medium[16,17] and laser cooling in vacuum[18]. However, the deliberate arrangement where a laser modifies a material's optical properties or position so that the material modifies the laser beam in return, constituting a nonlinear feedback mechanism, has been largely overlooked. We recently employed such arrangements to create laser-induced spatial nanostructures on various material surfaces with unprecedented uniformity[19], to produce 3D structures deep inside silicon[20], to demonstrate a new mechanism of highly-efficient laser material ablation[21], and to obtain complex behavior from dynamic self-assembly of colloidal nanoparticles[22].

Here, we demonstrate that we can dramatically enhance optical tweezers action by taking advantage of intracavity nonlinear feedback forces emerging when the microparticle is placed within the laser cavity, effectively implementing a closed-loop control.

We now describe an intracavity optical tweezers based on non-linear feedback forces using a ring-cavity fiber laser and a very low numerical aperture lens (NA = 0.12), which does not even permit trapping with a standard optical tweezers because of the overwhelming strength of the scattering forces that push the particle along the beam direction. We achieve intracavity optical trapping inside an active laser cavity where the laser mode is directly influenced by the position of the particle. We show that these optical tweezers can stably hold microscopic objects (3–7-μm-diameter polystyrene and silica particles), thanks to a power self-regulation, due to optomechanical coupling, obtaining a two-order-of-magnitude reduction of the average light intensity at the sample and the associated potential photodamage when compared with a standard optical tweezer that achieves the same degree of confinement (see Supplementary Fig. 1 and Supplementary Note 1).

## Results

**Working principle of intracavity optical trapping.** In our scheme, the particle is trapped inside the cavity of a fiber laser, which leads to a feedback between the position of the particle and the laser emission. When no particle is trapped (Fig. 1a), the optical loss of the cavity is low, and the laser operates above threshold, creating a strong optical potential. When a particle is trapped (Fig. 1b), the particle scatters light out of the cavity, increasing the cavity loss and reducing the quality factor of the cavity; this leads to an increase of the lasing threshold so that the laser turns off. When the particle tries to escape from the trap due to thermal fluctuations (Fig. 1c), it scatters less light so that the lasing threshold decreases, the laser power increases and the particle is pulled back. This optomechanical coupling between the trapped particle and the laser cavity leads to a self-regulation of the laser power, which increases whenever the particle is about to escape, and therefore permits us to stably hold micro-objects with a very low numerical aperture lens and at a low average intensity.

**Toy model.** We first describe a simple toy model for our intracavity trapping scheme to clarify how the nonlinear feedback forces emerge as a result of the interplay between the particle's motion and the laser's dynamics. Like any toy model, it is designed to be simplistic, while explaining clearly and concisely

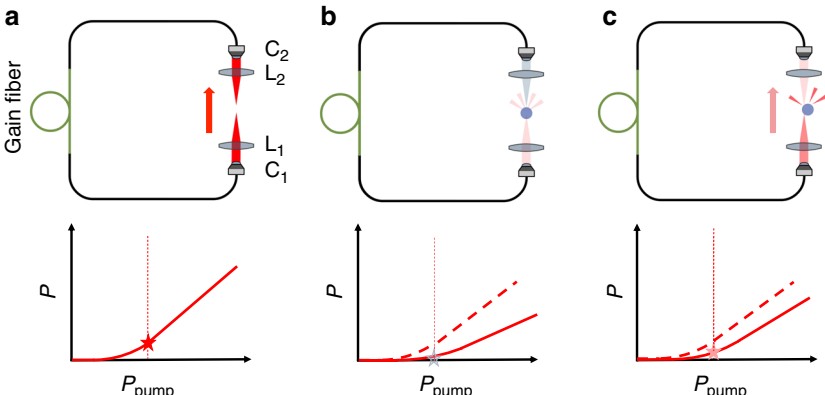

**Fig. 1** Intracavity optical trapping. The trapping optics (collimators $C_1$ and $C_2$, lenses $L_1$ and $L_2$) are placed within the cavity of a ring fiber laser (whose direction is indicated by the red arrows) so that the position of the particle can influence the cavity loss. **a** When the particle is not in the trap region, the optical loss of the cavity is low, the intracavity laser power $P$ is high, and consequently the particle is attracted toward the center of the trap. The laser power scaling curve (solid line) shows that the pump power $P_{pump}$ (vertical dashed line) is above the lasing threshold. **b** When the particle is at the center of the trap region, cavity losses due to scattering of light out of the cavity by the particle are maximum. The power scaling curve is right-shifted and the laser is below or barely above threshold for the same $P_{pump}$. The particle is not strongly trapped. **c** When thermal fluctuations displace the particle away from the trap region, the optical loss of the cavity decreases, $P$ increases, and the particle is pulled back toward the center of the trap

the nonlinear feedback mechanism underlying the trap. It also quantifies how and to what extent this scheme reduces the average laser power to which a trapped particle is exposed. For algebraic simplicity, we discuss motion in one dimension; generalization to three dimensions is straightforward, and the results are qualitatively the same. Here, we provide an overview, while we refer to the 'Methods' section for the details.

The motion of a Brownian particle suspended in a liquid medium and held in an attractive harmonic potential is described by the overdamped Langevin equation[3]

$$\dot{r}(t) = -\frac{k}{\gamma}r(t) + \sqrt{2D}\xi(t), \tag{1}$$

where $r(t)$ is the particle displacement from its equilibrium position, $k$ is the trap stiffness, $\gamma$ is the particle friction coefficient, $D$ is the particle diffusion coefficient, and $\xi(t)$ is a stochastic term corresponding to white noise with zero mean and unit power. The variance of the particle position in the trap is given by

$$\sigma_{r,OT}^2 = \frac{k_B T}{k}, \tag{2}$$

where $k_B$ is the Boltzmann's constant and $T$ is the ambient temperature. The trap stiffness is proportional to the laser power,

$$k = \kappa_P P, \tag{3}$$

where $\kappa_P$ is a proportionality constant determined by the geometrical and optical properties of the setup and the sample, but independent of the laser power. For standard optical tweezers, $P$ and $k$ are independent of $r$.

In the intracavity optical trapping scheme, the crucial difference is that the optically trapped particle is part of the laser cavity and the particle's position modulates the loss of the cavity. We describe this coupling using a well-known model for laser dynamics by H. Haken[23]. We choose the laser parameters such that the net gain of the laser is negative for particle displacements smaller than a finite amount, i.e., for $r < r_L$, so that the laser remains off. Once the particle reaches beyond $r_L$, the laser power turns on, increasing quadratically with $r$. Importantly, the timescale for the displacement of the particle (milliseconds) is much greater than the response time of the laser (nanoseconds), so that we can consider the laser to be always at its steady state for what concerns its effect on the particle motion. Therefore, the stationary value of the laser power is

$$P(r) = \begin{cases} 0 & r \leq r_L \\ P_0\left(\frac{r^2}{r_L^2} - 1\right) & r > r_L \end{cases} \tag{4}$$

where $P_0$ is a proportionality constant. $P(r)$ is plotted by the dashed red line in Fig. 2a for typical values of the parameters; the actual laser power saturates (solid line in Fig. 2a), but this occurs in a region where the particle probability density is negligible and therefore is not included in the toy model. The corresponding results from detailed simulations of the intracavity optical trapping (see below) show a similar nonlinear behavior (Fig. 2b).

Using Eq. (3), the restoring force is $F(r) = -kr = -\kappa_P P(r)r$, which is now non-harmonic because the trap stiffness depends on $r$ in the intracavity optical trapping scheme—this positional dependency constitutes the nonlinear feedback force, which permits this approach to go beyond what can be achieved with linear feedback. Integrating the force, we obtain the corresponding trap potential $U(r) = -\int_0^r F(x)dx$, and using the Boltzmann factor, the probability density of the particle position becomes

$$\rho(r) = \rho_0 e^{-\frac{U(r)}{k_B T}} = \begin{cases} \rho_0 & r \leq r_L \\ \rho_0 e^{-ar^4 + br^2 - \frac{1}{2}br_L^2} & r > r_L \end{cases} \tag{5}$$

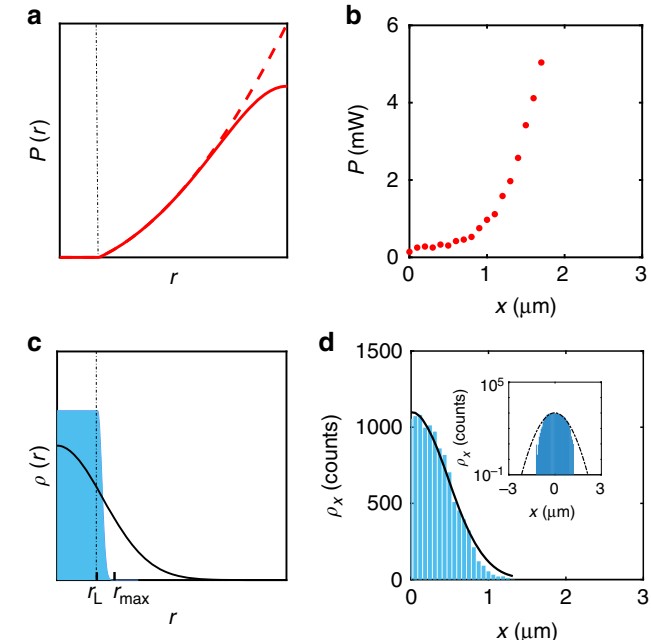

**Fig. 2** Dependence of the laser power on the particle position. **a** Laser power $P(r)$ as a function of particle position $r$ (Eq. (4)) employed in the toy model (dashed line) and the the actual laser power including saturation (solid line). **b** Intracavity laser power versus position for a 4.9-μm-diameter polystyrene particle obtained from detailed simulations of the intracavity optical trapping. **c** Corresponding probability density of the particle position (Eq. (5), with parameters $P_0 = 3$ mW, $r_L = 0.5$ μm, $\kappa_P = 0.1$ pN μm$^{-1}$ mW$^{-1}$, and $T = 300$ K). The solid line represents the probability density of the particle position obtained with a standard optical tweezer employing the same average power. **d** Probability distribution along the transverse x-direction obtained from detailed simulations for a 4.9-μm-diameter polystyrene particle held in the intracavity optical trap. The inset shows that the probability distribution is Gaussian (black line) for small displacements but it is sub-Gaussian for large displacement, where the nonlinear effect plays a leading role

where $a = \frac{P_0\kappa_P}{4r_L^2 k_B T}$, $b = \frac{P_0\kappa_P}{2k_B T}$, and $\rho_0$ is the normalization factor. The blue histogram in Fig. 2c shows an example of this probability distribution for typical values of the parameters. The corresponding histogram from detailed simulations of the intracavity optical trap (see below) is shown in Fig. 2d, where it can be clearly seen that the probability distribution is Gaussian for small displacements but becomes sub-Gaussian for large displacement because of the presence of the nonlinear feedback.

A key advantage of this toy model is that it can be solved exactly, obtaining expressions for the variance of the particle position, $\sigma_r^2 = \int_0^\infty r^2\rho(r)dr$, and the average laser power to which the particle is exposed, $P_{ave} = \int_0^\infty P(r)\rho(r)dr$ (these exact solutions, which are rather complex, are explicitly provided in the 'Methods' section). For example, using the values of the parameters employed in Fig. 2, we obtain $\sigma_r^2 = 0.11$ μm$^2$ and $P_{ave} = 0.09$ mW.

We can now compare these results in the case of a standard optical tweezer with the same average power. Using Eqs. (2) and (3), we obtain that $\sigma_{r,OT}^2 = \frac{k_B T}{\kappa_P P_{ave}}$; for the values of the parameters employed in Fig. 2, this corresponds to $\sigma_{r,OT}^2 = 0.36$ μm$^2$, which represents more than three times less confinement than in the intracavity optical trapping scheme. The corresponding probability distribution is shown by the solid black line plotted in Fig. 2c.

Further insight can be gained by analyzing some limiting cases that are amenable to simple analytical solutions. We will briefly consider a lower and an upper limit to the laser power exposure, providing the detailed derivation in the 'Methods' section. We first obtain a lower limit to the average laser power by neglecting the $r^2$ term in the Maxwell-Boltzmann distribution (Eq. (5)) so that the probability function drops faster than in the exact case. This yields

$$P_{\mathrm{ave}}^{\mathrm{L}} \simeq \frac{2}{P_0}\left(\frac{k_{\mathrm{B}}T}{\kappa_{\mathrm{p}}r_{\mathrm{L}}^2}\right)^2, \qquad (6)$$

which corresponds to $P_{\mathrm{ave}}^{\mathrm{L}} = 0.02\,\mathrm{mW}$ for the parameters employed above. Next, we obtain an upper limit by considering a uniform distribution extending to some $r_{\max}$, where $\rho(r_{\max}) \ll 1/\mathcal{C}$ with $\mathcal{C} \gg 1$ (the result is highly insensitive to the particular choice of $r_{\max}$):

$$P_{\mathrm{ave}}^{\mathrm{U}} \simeq P(r_{\max})\sqrt{\ln(\mathcal{C})\frac{k_{\mathrm{B}}T}{4r_{\mathrm{L}}^2\kappa_{\mathrm{p}}P_0}}, \qquad (7)$$

which corresponds to $P_{\mathrm{ave}}^{\mathrm{U}} = 0.70\,\mathrm{mW}$ for the parameters employed above. These results show that the average power exposure is much reduced, if the laser can turn on sharply (i.e., large $P_0$ or large small-signal gain); this result motivates the choice of a fiber laser, because these lasers have extremely high small-signal gain factors (30–40 dB)[24]. Furthermore, this shows that, from an experimental perspective, we are most interested in the case of $r_{\mathrm{L}} \lesssim r_{\max}$. For $r < r_{\mathrm{L}}$, the laser is off and the particle experiences free diffusion with a uniform probability density. To take advantage of intracavity trapping, we are interested in parameter combinations for which free diffusion is dominant. In this case, the laser has to be off or operating at low power up to some large $r_{\mathrm{L}}$ and be turned on quickly for $r > r_{\mathrm{L}}$, so as to erect a steep potential barrier that confines the particle.

**Simulations**. The toy model we have discussed until now is a simplified model that describes the qualitative behavior of the intracavity optical trapping scheme as well as its nonlinear response for large particle displacements. In this model, the power and hence trapping force are considered to be zero for small particle displacements. However, in reality they have small values that do operate the trap even when the particle is near the equilibrium position. Thus, we need an accurate description of the coupling between the laser and the trapped particle thermal dynamics at equilibrium to compare with experiments. In particular, accurate simulations can help to associate an effective harmonic potential to the optical trap for small displacements from the equilibrium position, and hence to define a meaningful stiffness using the standard calibration methods based on the thermal fluctuations of a trapped particle (see Supplementary Figs. 2–4, and Supplementary Notes 2 and 3).

We therefore present a series of numerical simulations based on an extended theoretical model, including highly realistic descriptions of the laser dynamics, optical losses incurred by the particle, and the particle's Brownian motion in order to gain a quantitative understanding of the dynamics of intracavity optical trapping and to guide the experiments. The dynamics of the trapped particle are similarly governed by the interplay between the optical force $\mathbf{F}_{\mathrm{ot}}(\mathbf{r}, P)$, the gravity minus the buoyancy $\mathbf{F}_{\mathrm{g}}$, the viscous drag acting on the particle, and the thermal fluctuations. The resulting overdamped Langevin equation is[3]:

$$\dot{\mathbf{r}} = \gamma^{-1}\left[\mathbf{F}_{\mathrm{ot}}(\mathbf{r}, P) + \mathbf{F}_{\mathrm{g}}\right] + \sqrt{2D}\mathbf{W}(t), \qquad (8)$$

where $\mathbf{r}$ is the particle position, $\gamma = 6\pi\eta R$ is its friction coefficient that depends on the particle radius $R$ and the medium viscosity $\eta$, $P$ is the power of the laser, $D = k_{\mathrm{B}}T/\gamma$ is the particle diffusion that depends on the temperature $T$, $k_{\mathrm{B}}$ is the Boltzmann constant, and $\mathbf{W}(t)$ is a vector of independent white noises.

The laser dynamics is modeled using standard power rate equation[24] (see 'Methods' section for details). This is a highly realistic model that includes gain saturation, which was ignored in the toy model. We note that the characteristic timescale for particle displacement due to Brownian motion is in the millisecond range, whereas the laser dynamics is in the nanosecond range. Thus, we take the laser to be always at its steady state and calculate its power, given the loss corresponding to the particle position.

Since the size of the particles is significantly larger than the wavelength[25], we have calculated the trapping force, as well as the loss, using the geometrical optics approach implemented in the software package Optical Tweezers in Geometrical Optics (OTGO)[26]: The incoming laser beam is decomposed into a set of optical rays, which are then focused by the focusing lens. As the rays reach the particle, they get partially reflected and partially transmitted. The direction of the reflected and transmitted rays are different from those of the incoming rays. This change of direction entails a change of momentum and a force acts on the particle due to the action–reaction law. If the refractive index of the particle is greater than that of the medium as is usually the case, these optical forces tend to pull the sphere toward a stable equilibrium position near the focal spot. The optical forces are proportional to the laser power. As the scattered rays reach the collecting lens, they are collected and projected onto the back-focal plane input of the fiber.

Figure 3 illustrates the simulations' results for a 4.9-μm-diameter polystyrene particle held in an intracavity optical trap. The particle confinement is $\sigma_r^2 = 0.18\ \mu\mathrm{m}^2$ and $\sigma_z^2 = 0.38\ \mu\mathrm{m}^2$, which are in good agreement with the results obtained for the experiments presented in the next section. Some interesting configurations of particle and laser are shown in Fig. 3a, corresponding to the positions indicated in Fig. 3b. When the particle is in the center of the trap, a significant part of the light is scattered out of the collector lens, as can be seen in the ray-optics diagram shown in panel i of Fig. 3a, which leads to a low laser power (position i in Fig. 3b). When the particle moves away from the trap along the radial direction (panel ii of Fig. 3a), this leads to an increase of the laser power that reaches the collector lens (position ii in Fig. 3b) and, therefore, to an increased restoring force pulling back the particle. Similarly, when the particle moves down along the axial direction (panel iii of Fig. 3a), there is an increase of the collected power (position iii in Fig. 3b) and an increase of the scattering force that pushes the particle up. Finally, when the particle moves up along the axial direction (panel iv of Fig. 3a), the rays are scattered away from the collector lens (position iv in Fig. 3e) and this decreases the laser power letting the particle fall back toward the center of the trap.

To better understand the coupling between the particle position fluctuations and the laser power, we have plotted the laser power as a function of the radial and axial position of the particle in Fig. 3c. The power increases whenever the radial position $r$ increases, confirming that, when the particle tries to escape along the radial direction, the laser increases its power and pulls back the particle toward the center of the trap. When the particle moves down, the laser power increases and the scattering force pushes the particle up toward the trap center, and, when the particle moves up, the laser power and associated scattering force decrease, which permits the

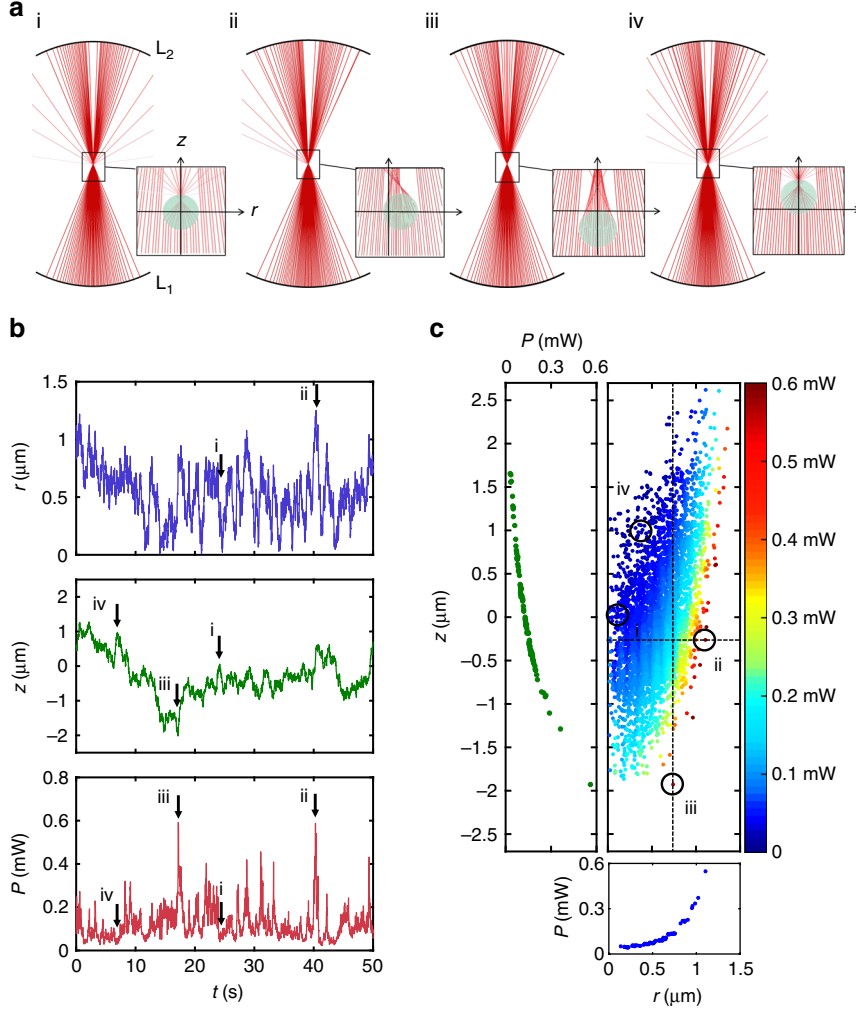

**Fig. 3** Detailed simulations. **a** Ray-optics diagrams of the propagation of a focused beam through an optically trapped particle (i) when the particle is at the center of the trap (equilibrium position in the trap that takes into account also the effective gravitational force acting on the particle), (ii) when it is displaced in the radial direction, (iii) when it is displaced along the axial direction downward and (iv) upward. **b** Radial ($r$) and axial ($z$) particle position, and corresponding laser power ($P$) obtained from the simulation of the motion of a 4.9-µm-diameter polystyrene particle trapped in the intracavity optical trap. **c** Dependence of the laser power on the radial and axial position of the particle. The points (i)–(iv) correspond to the configurations in (**a**) and the dashed lines correspond to the insets graphs the dependence on $z$ and $r$ on the left and bottom, respectively

particle to fall down toward the center of the trap. These results can be directly compared with the toy model results shown in Fig. 2. We observe that for small displacements the power is small (Fig. 2b), rather than zero, and constant (Fig. 2b, Supplementary Fig. 4 and Supplementary Note 3), and the force is linear consistent with a Hookean response of the trap (Supplementary Fig. 4 and Supplementary Note 3). In fact, the power is below the laser threshold in this linear regime for small displacements and increases suddenly for large displacements within the nonlinear regime. As a consequence, the corresponding probability distribution (Fig. 2d) is close to a Gaussian enabling a calibration of optical forces for small displacements (Supplementary Fig. 2 and Supplementary Note 2). For large displacements nonlinear effects can be observed in the tails that yield a sub-Gaussian probability distribution (inset of Fig. 2d). Furthermore, the positional fluctuations follow the Gaussian distribution typical of an optical tweezer, which for small displacements is well approximated with a harmonic potential.

Differently, for high-NA optical tweezers the power is decoupled from the trapped particle fluctuations (Supplementary

Note 4). High-NA optical tweezers require a minimum power that is higher than an intracavity optical trap in order to achieve stable trapping. Clearly, this power is constant as a function of the particle position (Supplementary Fig. 5).

We note that the basic theoretical concepts hold also for particles smaller than the trap wavelength. However for our experimental parameters, the smaller the particle the less effective the intracavity trapping. In fact, it is possible to describe the light-particle interaction and optical forces in intracavity optical trapping of small particles by exploiting the dipole approximation[3,7] (see Supplementary Note 5). Since for small particles the power scattered in the trap decreases rapidly, the nonlinear feedback that regulates intracavity trapping is reduced. Thus, for negligible optical losses, the intracavity trap behaves as a standard single-beam optical tweezer and cannot efficiently trap small particle at low intensity (see Supplementary Fig. 6). For our experimental parameters, microparticles are needed to increase light scattering and hence optical losses in the trap to efficiently operate the intracavity feedback trapping. Thus, the intracavity feedback trapping is efficient at the microscale while reduces to a standard single-beam optical trapping at the nanoscale that

cannot efficiently trap nanoparticles. However, the intracavity optical trapping approach can in principle be scaled down to particles significantly smaller than the wavelength by changing the experimental parameters. In particular, increasing the numerical aperture would allow to increase the losses due to the particle and therefore make the intracavity optical trapping more efficient at the nanoscale.

**Experimental results**. Finally, guided by the simulation results, we have built an experimental setup to prove the operational principle of intracavity optical trapping (Fig. 4a). We constructed a continuous-wave ring-cavity fiber laser emitting at 1030 nm with the trapping optics placed inside the cavity. The laser beam is directed upward and the trap is achieved by a lens with an effective $NA = w/f = 0.12$, where $w = 1.0$ mm is the beam waist at the lens and $f = 8.0$ mm is the lens focal length.

The cavity further comprises a single-mode Yb-doped fiber (Yb 1200-6/125, nLIGHT, Inc., core diameter of 6 μm, cladding diameter of 125 μm) as gain medium, pumped by a single-mode fiber-pigtailed diode-laser at 976 nm through a wavelength division multiplexer. To achieve spectral stability, we included a fiberized band-pass filter (centered at 1030 nm, full width at half maximum of 2 nm, placed after the gain fiber). After the band-pass filter, the beam is split by a coupler with 99:1 coupling ratio: the larger portion is sent to a single-mode fiber-pigtailed collimator (OZ Optics, Ltd.), and the smaller portion is sent to a photodiode power sensor (S150C, Thorlabs, Inc., 10 pW resolution) for power monitoring. The output of the collimator is reflected by a short-pass dichroic mirror (DMSP1000, Thorlabs, Inc.) and focused on the sample by an aspheric lens (8.0 mm focal length, NA = 0.12). The laser light is then collected by a second identical aspheric lens, reflected by a short-pass dichroic mirror (DMSP1000, Thorlabs, Inc.), and coupled back into the fiber by a collimator. An in-line isolator is used to ensure that the light is traveling within the cavity unidirectionally. The response time of the laser, as measured using an acousto-optic modulator inside the cavity, is about 20 ns (see Supplementary Note 6). This is several orders of magnitude faster than the dynamics of the

intracavity trapping (~100 ms, see Supplementary Fig. 7) and the Brownian dynamics at equilibrium (~10 s) that shows up in the autocorrelation function and power spectrum analysis (see Supplementary Figs. 2 and 3) of the trapped particle tracking signals (see Supplementary Note 2). The red circles in Fig. 4b show the power scaling curve of the laser as a function of the pump power.

We have suspended polystyrene (diameter of 4.9 and 6.2 μm, Microparticles, GmbH) or silica (diameter of 2.8, 4.0, and 4.8 μm, Microparticles, GmbH) particles in water and placed a droplet of the resulting solution in a sample chamber realized between two microscope slides separated by a parafilm layer (100-μm thick). We placed the sample on a 3-axis translation stage. The orange squares in Fig. 4b show the power scaling curve of the laser as a function of the pump power when a 4.9-μm-diameter polystyrene particle placed at focal point: at the pump power of 66 mW, the laser power is reduced from 5.0 to 0.2 mW.

For imaging, we have illuminated the sample using a LED (630 nm wavelength, 20 nm bandwidth) whose coherence length is much shorter than the thickness of the microscope slides and the separation between the two slides to avoid interference. We have recorded the motion of the particle using a CMOS camera (DCC1645C, Thorlabs, Inc., 50 Hz frame rate) and used digital video microscopy to track its trajectory in 3D (see 'Methods' section).

In Fig. 5, we show the results we obtained by trapping a 4.9-μm-diameter polystyrene particle at a pump power of 66 mW corresponding to 5.0 mW laser output in the absence of the particle. Figure 5a shows the time evolution of the particle radial position $r$, its axial position $z$, and the laser power $P$. We have measured the confinement achieved by the trap in the radial and axial directions calculating the variance of the particle position, which are $\sigma_r^2 = 0.038 \mu m^2$ and $\sigma_z^2 = 0.41 \mu m^2$, whose numerical values are in good agreement with the results of the simulations presented in the previous section. Figure 5b shows how the laser power depends on the radial and axial particle position, which is also in agreement with the results of the simulations presented in Fig. 3c. Along the radial direction, the laser power increases when $r$ is large, providing enough

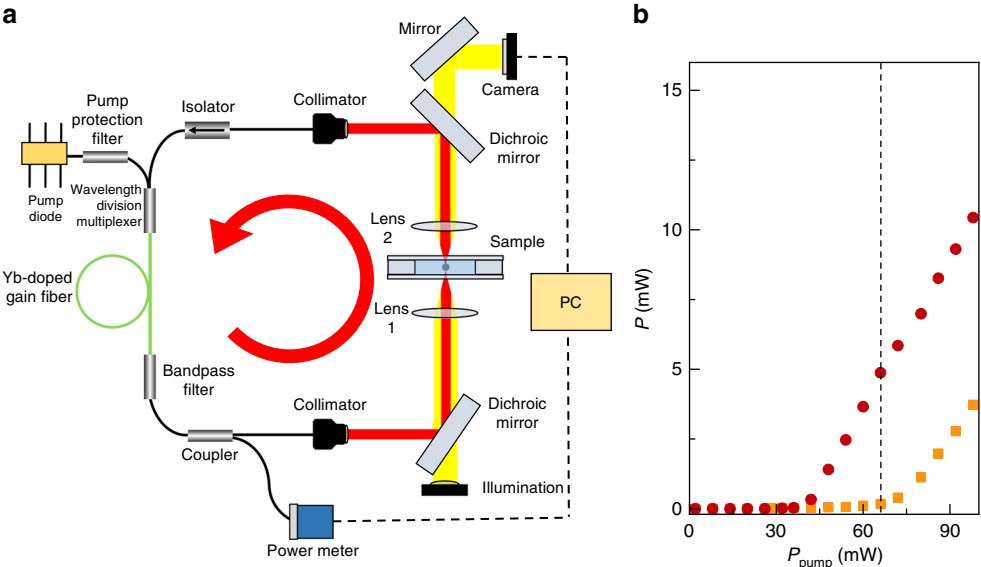

**Fig. 4** Experimental setup. **a** The setup comprises a diode-pumped Yb-doped fiber laser, the trapping optics, and the digital video microscope. The arrow represents the direction in which the light travels. **b** Measured power scaling with a trapped 4.9-μm-diameter polystyrene particle (orange squares) and without the trapped particle (red circles). At a pump power of 66 mW (dashed vertical line), the laser is below threshold with the particle (orange squares), but above threshold without the particle (red circles)

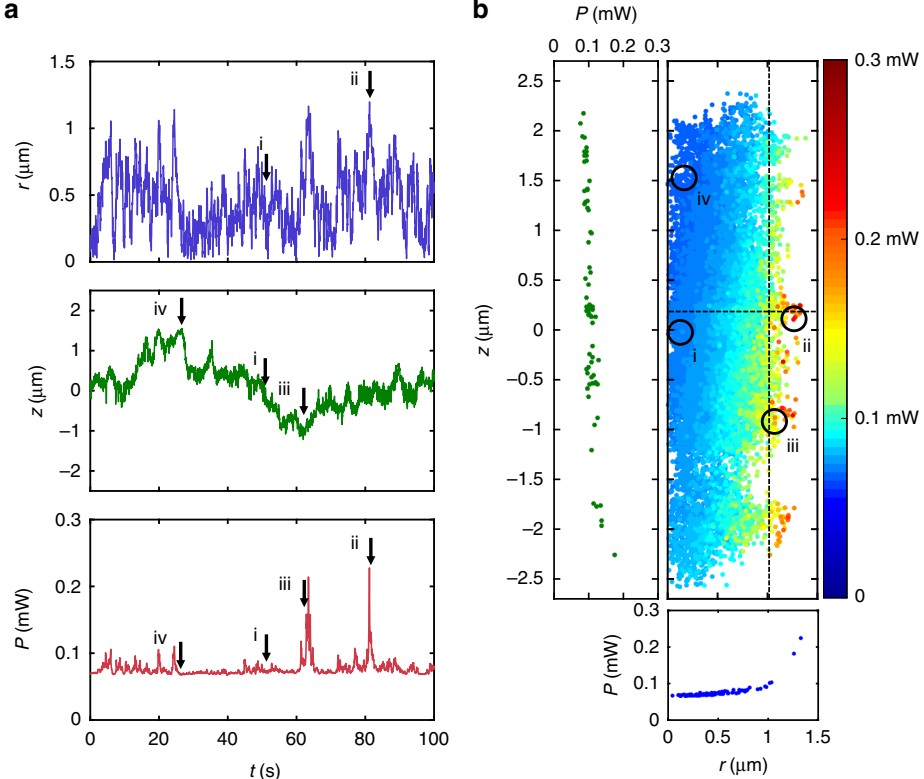

**Fig. 5** Experiments. **a** Radial ($r$) and axial ($z$) particle position, and corresponding laser power ($P$) for a 4.9-μm-diameter polystyrene held in the intracavity optical trap. **b** Dependence of the laser power on the radial and axial position of the particle. The positions indicated with i, ii, iii, and iv correspond to particle position and laser power configurations similar to those illustrated in the diagrams shown in Fig. 3a. The points (i)–(iv) correspond to the configurations shown in Fig. 3a, and the dashed lines correspond to the insets graphs the dependence on $z$ and $r$ on the left and bottom, respectively

restoring force to push the particle back toward the center of the trap. Along the axial direction, the laser power decreases when the particle moves upward leading to the particle moving downward because of sedimentation, and increases when the particle moves downward, leading to enhanced scattering forces pushing the particle back upward.

**Comparison with standard optical tweezers**. It is instructive to compare intracavity optical trapping with a standard single-beam optical tweezer. To this end, we have compared the inverse radial and axial confinement, $\sigma_r^{-2}$ and $\sigma_z^{-2}$, per unit intensity at the sample, $I$, measured using the intracavity optical tweezers to those obtained using a different setup with low and high numerical aperture lenses where intensity is constant and no feedback is occurring.

For the case of low-NA standard optical tweezers (NA = 0.12), we observe that the particle is never trapped along the axial direction because the restoring force is not strong enough. Therefore, intracavity optical trapping provides a simple, self-aligning technique to achieve trapping at low NA. We remark that several alternative approaches have been proposed to make optical tweezers capable of trapping particles at low NA (see the comparison in Supplementary Fig. 1 and Supplementary Note 1). Clearly, each technique has its own strengths and weaknesses with respect to intracavity trapping. For example many of these methods require the use of multiple or structured optical beams, such as in counterpropagating optical tweezers[27–30], in mirror trapping[31] and in trapping with focused Bessel beams[32], or special sample preparation, such as in trapping using self-induced back action[16,17]. Intracavity trapping has fundamental differences in that it operates at the microscale with an all-optical nonlinear

feedback scheme coupling the laser cavity with the optomechanical response of the trapped particle.

For the case of high-NA standard optical tweezers (NA = 1.30), we trapped a 4.9 μm-diameter polystyrene particle with a standard optical tweezer using a high-NA microscope objective (NA = 1.30) using a laser with a central wavelength of 976 nm. We measured the particle trajectories using digital video microscopy and analyzed them to obtain the trap stiffness. We measured $\sigma_{hNA,r}^{-2}/I_{hNA} = 6.3\,mW^{-1}$ and $\sigma_{hNA,z}^{-2}/I_{hNA} = 3.7\ mW^{-1}$, where $I_{hNA} = 35\ mW\,\mu m^{-2}$ is the intensity at the sample. Instead, the intracavity optical trap achieves about two order of magnitude tighter confinement for the same intensity at the sample: $\sigma_r^{-2}/I = 2180\,mW^{-1}$ and $\sigma_z^{-2}/I = 200\,mW^{-1}$, where $I = 0.012\ \mu m\,Wm^{-2}$ is the intensity at the sample.

Figure 6 shows a comparison of the experimental and simulated confinement per unit intensity at the sample for polystyrene and silica particles of various sizes. This is a quantity that is directly related to the effective stiffness of the trapping potential explored by the particle for small displacement. The intracavity confinement per unit intensity is consistently about two orders of magnitude higher than that for standard optical tweezers. We observe that for the radial direction we find a good agreement with no free parameters between experimental and simulated results. On the other hand, for the axial direction the intracavity experimental confinement is smaller than the simulated curves. This can be accounted for by noting that simulations do not take into account effects such as aberrations or distortions by the sample chamber (glass-water interfaces) that might change the feedback and trapping point, weakening the axial trapping efficiency in experiments. Furthermore, for the axial direction ray optics might not provide a very accurate

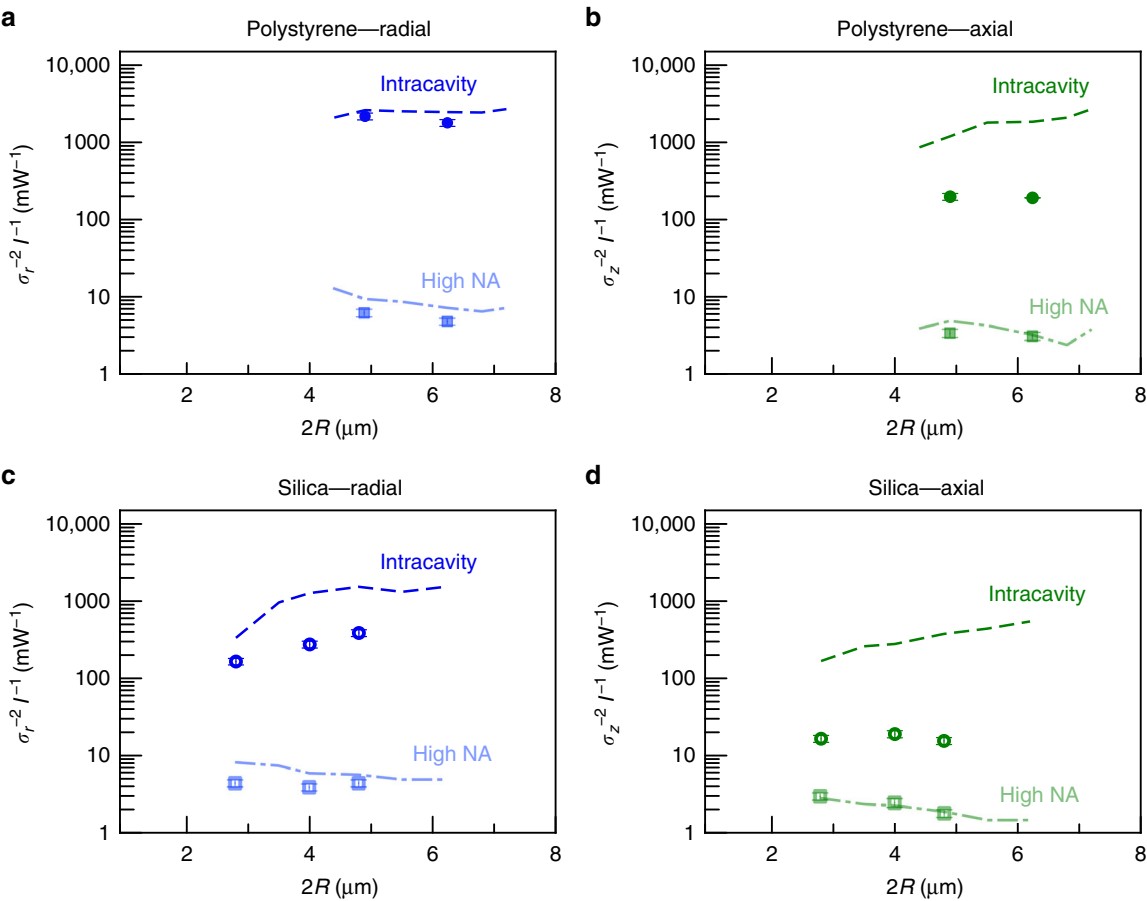

**Fig. 6** Enhancement of particle confinement per unit intensity at the sample. Comparison of the experimentally measured inverse radial and axial trap confinement ($\sigma_r^{-2}$ and $\sigma_z^{-2}$, respectively) per unit intensity at the sample for an intracavity optical trap (circles) and a standard high-NA optical tweezer (squares) for polystyrene **a**, **b** and silica **c**, **d** particles of various radii $R$. The dashed lines are the corresponding results from numerical simulations with no free parameters. In the radial direction the agreement between experiments and simulations is quite good, while in the axial direction the experimental results are smaller than the simulated ones because in this direction aberration and distortion from the glass-water interfaces can play an important role in weakening the feedback and axial trapping stiffness in the intracavity trap

description of optical forces even for this large particle size[33,34]. Therefore, we can consider that the overall agreement between our ray-optics theory without free parameters and experiments to be quite fair.

## Discussion

We have demonstrated through a simple analytical model, simulations of a realistic model, and experiments, a novel optical trapping scheme where the laser operation is nonlinearly coupled to the motion of the trapped particle. This coupling gives rise to instrinsic nonlinear feedback forces that confine microparticles efficiently at low intensity. We implemented this scheme within the cavity of a fiber laser because its large, small-signal gain is essential for achieving a high trap stiffness per unit intensity. We have trapped polystyrene and silica microparticles of different sizes. Tracking of the particle thermal fluctuations has enabled us to investigate the optomechanical coupling between the ring-cavity fiber laser and the optical trap elucidating its working principles. For small displacements from the equilibrium position, the intracavity trap is well approximated by an effective harmonic potential that can be calibrated by standard methods[8] (see Supplementary Note 2) yielding a direct calibration of the trap stiffness. Thus, intracavity optical trapping can also be used to controllably apply and measure small forces.

One of the major advantages of the intracavity optical trapping scheme is that it can operate with very low-NA lenses, with a consequent large field-of-view, and at very low average power, resulting in about two orders of magnitude reduction in exposure to laser intensity compared with standard optical tweezers. When compared with other low-NA optical trapping schemes such as SIBA trapping[16,17], counterpropagating beam[28–30,35] or mirror optical trapping[31], positive and negative aspects can be considered, such as in terms of trap stiffness and average irradiance of the sample (see Supplementary Fig. 1 and Supplementary Note 1). Another advantage of the intracavity optical trapping scheme is that it intrinsically features a high bandwidth thanks to the intracavity feedback. Furthermore, since intracavity optical trapping is a self-regulated mechanism, it can be designed to work for different particle types without the need to explicitly determine their properties, such as in any external feedback scheme which requires an explicit detection and identification of the particle and its properties. In summary, intracavity optical trapping is all-optical and easy to implement as the feedback mechanism is intrinsically built in; furthermore, it has no need for separate electronics or recalibration. Thus, intracavity optical trapping is fundamentally different from other approaches as it is an all-optical nonlinear feedback scheme that exploits a single-beam in a laser cavity.

These features can yield advantages when dealing with biological samples. In fact, biological matter is sensitive to light intensity and the typical tight focus of standard optical tweezers has detrimental effects over cell manipulation, phototoxicity, and

long-term survival. Ultra-low intensity at our wavelength can grant a safe, temperature controlled environment, away from surfaces for microfluidics manipulation of biosamples. There are multiple mechanisms that can yield cell cycle inhibition and destruction in optical trapping that are related with local heating[36,37], induction of reactive oxygen species, light-induced protein inactivation, and specific pigment absorption[38]. Accurate studies on *Saccharomices cerevisiae* yeast cells in near-infrared counterpropagating traps[39] and standard optical tweezers[40] have found no evidence for a lower power threshold for phototoxicity. In particular, in counterpropagating traps[39] it has been shown that 3.5 mW power (corresponding to an intensity of about 0.33 mW $\mu$m$^{-2}$) is needed for 3D trapping of a single cell, but 0.7 mW (corresponding to about 0.07 mW $\mu$m$^{-2}$) is already sufficient to detect phototoxicity over long-term exposure time (few hours). In standard optical tweezers[40], a power of 10 mW (about 22 mW $\mu$m$^{-2}$ intensity) has been reported as the minimum 3D trapping power, but a pulsed mode operation of optical tweezers has been observed to reduce phototoxicity because of the reduced interaction area. Despite the amount of sustainable dosage by a cell or specific mechanisms involved are still debatable[40], we observed that we can 3D trap single yeast cells with about 0.47 mW, corresponding to an intensity of 0.036 mW $\mu$m$^{-2}$, that is more than a tenfold less intensity than standard techniques.

We anticipate that intracavity optical traps, enabling 3D confinement with long-working-distance lenses, low operating intensity, simple optics, and low costs can prove useful in several research fields, particularly in biology where low photodamage of the sample is often crucial. It is also tantalizing to consider the interplay that might arise when the timescales of the laser and of the particle dynamics become comparable, e.g., when trapping particles in air or in vacuum.

## Methods

**Detailed description of the toy model.** First, we derive the model for laser dynamics in Eq. (4). We describe the laser dynamics using a model based on the one introduced by H. Haken[23]

$$\dot{P} = NWP - \frac{l(r)}{\tau_R}P, \tag{9}$$

where $N = N_0 - \frac{2N_0\tau}{h\nu}P$ is the electron population in the excited state, $N_0$ is the population in the excited state determined by the pump power, $W$ is the spontaneous emission rate at which excited electrons relax to the ground state, $\tau$ is the relaxation time, $\tau_R$ is the cavity round-trip time, $h$ is the Planck's constant, $\nu$ is the optical frequency of the laser output, and $l(r)$ is the loss of the cavity, which depends on the particle position $r$. Inserting $N$ into the equation above gives

$$\dot{P} = G_{net}(r)P - \frac{2N_0\tau W}{h\nu}P^2, \tag{10}$$

where $G_{net}(r) = N_0 W - l(r)/\tau_R$ is the net gain, which depends on the particle position. While the dependence of the losses on the particle's position is, in general, complex, it is maximum at the center and drops quadratically for small displacements as $l(r) = l_0(1 - r^2/r_c^2)$, where $r_c$ characterizes the scaling of the losses with the particle's position, which depends on the radius of the particle, relative indices of refraction of the particle and the liquid medium, and the losses due to absorption and scattering. The laser cavity is intentionally arranged to be below threshold (i.e., $G_{net}(r) < 0$) for $r < r_L$, where $r_L$ is the characteristic particle displacement where lasing turns on. Therefore, the stationary value of the laser power is given by Eq. (4), i.e.,

$$P(r) = \begin{cases} 0 & r \leq r_L \\ P_0\left(\frac{r^2}{r_L^2} - 1\right) & r > r_L \end{cases}$$

where $G_0 = l_0/\tau_R - N_0 W$, $r_L = r_c\sqrt{G_0\tau_R/l_0}$, and $P_0 = h\nu G_0/(N_0\tau W)$, which is the the normalized slope of laser power with respect to the particle position and depends on laser parameters, such as optical loss, gain characteristics, and pump power. A more complete model would incorporate gain saturation as well as the saturation of the decrease of losses with increasing $r$, which ultimately limits the laser power to a finite value. Nevertheless, we can use this algebraically simpler model because the probability of the particle to be displaced to large values of $r$, where saturation matters, decreases exponentially.

Second, we derive the stationary probability density of the particle position in Eq. (5). The restoring force is given by

$$F(r) = -kr = -\kappa_P P(r)r = \begin{cases} 0 & r \leq r_L \\ -P_0\kappa_P\left(\frac{r^2}{r_L^2} - 1\right)r & r > r_L \end{cases} \tag{11}$$

This is an important result that warrants several comments. First, it is the dependence of trap stiffness on position that constitutes a nonlinear feedback mechanism. Second, the particular form of this nonlinear response is such that the laser power is zero up to a certain displacement, $r_L$, and increases quadratically afterward (until reaching saturation, which is ignored here). We will show below that this particular form results in a much reduced average power that a trapped particle experiences, compared with having fixed laser power as in a traditional trap for the same level of confinement (i.e., variance in the trap). Integrating the force, the corresponding trap potential is found to be

$$U(r) = -\int_0^r F(x)dx = \begin{cases} 0 & r \leq r_L \\ P_0\kappa_P\left(\frac{r^4}{4r_L^2} - \frac{r^2}{2} + \frac{r_L^2}{4}\right) & r > r_L \end{cases} \tag{12}$$

From the potential, we use the Boltzmann factor to determine the probability density of the particle position, obtaining Eq. (5).

Third, we derive the analytical expressions for $\sigma_r^2$ and $P_{ave}$. We can calculate the variance of the particle position as

$$\sigma_r^2 = \int_0^\infty r^2\rho(r)dr = \int_0^{r_L} r^2\rho_0 dr + \int_{r_L}^\infty r^2\rho_0 e^{-ar^4 + br^2 - br_L^2/2}dr. \tag{13}$$

and the average laser power value to which the particle is exposed as

$$P_{ave} = \int_0^\infty P(r)\rho(r)dr = \int_{r_L}^\infty P_0\left(\frac{r^2}{r_L^2} - 1\right)\rho_0 e^{-ar^4 + br^2 - br_L^2/2}dr. \tag{14}$$

The solutions of integrals in Eqs. (13) and (14) can be expressed as series: If we expand $e^{br^2}$ in a Taylor series, with its convergence guaranteed by the faster decaying term, $e^{-ar^4}$, we can exactly evaluate the integrals corresponding to each term using $\int_{r_L}^\infty r^{2n}e^{-ar^4}dr = \frac{\Gamma\left(\frac{2n+1}{4}, ar_L^4\right)}{4a^{\frac{2n+1}{4}}}$. Thus, the variance and the average power exposure are

$$\sigma_r^2 = \frac{\frac{r_L^3}{3} + e^{-br_L^2/2}\sum_{n=0}^\infty \frac{b^n\Gamma\left(\frac{2n+3}{4}, ar_L^4\right)}{4a^{\frac{2n+3}{4}}n!}}{r_L + e^{-br_L^2/2}\sum_{n=0}^\infty \left(\frac{P_0 b^n\Gamma\left(\frac{2n+1}{4}, ar_L^4\right)}{4a^{\frac{2n+1}{4}}n!}\right)} \tag{15}$$

and

$$P_{ave} = \frac{e^{-br_L^2/2}\sum_{n=0}^\infty \left(\frac{P_0 b^n\Gamma\left(\frac{2n+3}{4}, ar_L^4\right)}{4a^{\frac{2n+3}{4}}r_L^2 n!} - \frac{P_0 b^n\Gamma\left(\frac{2n+1}{4}, ar_L^4\right)}{4a^{\frac{2n+1}{4}}n!}\right)}{r_L + e^{-br_L^2/2}\sum_{n=0}^\infty \left(\frac{P_0 b^n\Gamma\left(\frac{2n+1}{4}, ar_L^4\right)}{4a^{\frac{2n+1}{4}}n!}\right)}, \tag{16}$$

where $\Gamma(s, x)$ is the upper incomplete gamma function defined as $\Gamma(s, x) = \int_x^\infty u^{s-1}e^{-u}du$[41].

Fourth, we derive the lower limit for the exposure to laser power (Eq. (6)). We can calculate a lower limit to the average laser power to which the particle is exposed by neglecting the $r^2$ term in the expression for $\rho(r)$ (Eq. (5)). Compared with the exact form, the probability function drops faster without the $r^2$ term, ensuring that such a calculation constitutes a lower limit. We therefore obtain

$$\sigma_{L,r}^2 \simeq \rho_0'\int_0^{r_L} r^2 dr + \rho_0'\int_{r_L}^\infty r^2 e^{-ar^4 + br_L^2/2}dr \tag{17}$$

and

$$P_{ave}^L \simeq \rho_0'\int_{r_L}^\infty P_0\left(\frac{r^2}{r_L^2} - 1\right)e^{-ar^4 + br_L^2/2}dr, \tag{18}$$

where $\rho_0' = \left[\int_0^{r_L} dr + \int_{r_L}^\infty e^{-ar^4 + br_L^2/2}dr\right]^{-1} = \left[r_L + \frac{e^{br_L^2/2}}{4a^{\frac{1}{4}}}\Gamma\left(\frac{1}{4}, ar_L^4\right)\right]^{-1}$. These integrals can be evaluated using the upper incomplete gamma functions, obtaining

$$\sigma_{L,r}^2 \simeq \rho_0'\left[\frac{r_L^3}{3} + \frac{e^{br_L^2/2}}{4a^{\frac{3}{4}}}\Gamma\left(\frac{3}{4}, ar_L^4\right)\right] \tag{19}$$

and

$$P_{ave}^L \simeq P_0\frac{\rho_0'}{4a^{\frac{1}{4}}}e^{br_L^2/2}\left[\frac{1}{\sqrt{ar_L^2}}\Gamma\left(\frac{3}{4}, ar_L^4\right) - \Gamma\left(\frac{1}{4}, ar_L^4\right)\right]. \tag{20}$$

Even though these results are also not completely transparent, it is evident from the equation above that $P_{ave}^L$ will be much smaller than $P_0$ for reasonable choices of the parameters because $\rho_0'$ is in the order of $r_L$, $a^{\frac{1}{4}}$ is much larger than $r_L$, and the remaining terms are either order of 1 or smaller. To estimate $P_{ave}^L$ for $ar_L^4 \gg 1$ and

$P_0 \kappa_P r_L^2 \gg 4 k_B T$, we can use the asymptotic series[41]

$$\Gamma(a, x) = x^{a-1} e^{-x} \left( 1 + \frac{a-1}{x} + \dots \right), \qquad (21)$$

which is valid for large $x$. Retaining the first two terms of the series, we obtain

$$\Gamma\left(\frac{3}{4}, ar_L^4\right) \simeq e^{-ar_L^4} \frac{1}{a^{1/4} r_L} \left( 1 - \frac{1}{4 ar_L^4} \right), \qquad (22)$$

$$\Gamma\left(\frac{1}{4}, ar_L^4\right) \simeq e^{-ar_L^4} \frac{1}{a^{3/4} r_L^3} \left( 1 - \frac{3}{4 ar_L^4} \right). \qquad (23)$$

The substitution of the above equations into Eq. (20) gives the lower bound for $P_{ave}$ in Eq. (6):

$$P_{ave}^L \simeq \frac{2/P_0}{1 + \frac{1}{4 ar_L^4}\left(1 - \frac{3}{4 ar_L^4}\right)} \left(\frac{k_B T}{\kappa_P r_L^2}\right)^2 \simeq \frac{2}{P_0}\left(\frac{k_B T}{\kappa_P r_L^2}\right)^2.$$

Finally, we derive the upper limit to exposure to laser power (Eq. (7)). To calculate an upper limit to the laser power, we assume a uniform distribution of the particle position. We make use of the fact that, while $\rho(z)$ extends to infinity, it drops extremely fast as a result of the quartic term of $r$ within the exponential. This creates an excellent opportunity for the calculation of an approximate result and suggests a well-defined point to terminate the uniform distribution. To this end, we introduce a new parameter, $r_{max}$, which represents a finite amount of displacement beyond which the probability of finding the particle is taken to be negligible. More precisely, at $r_{max}$, the probability is $1/\mathcal{C}$, where $\mathcal{C} \gg 1$. This value is given by $e^{-ar_{max}^4 + br_{max}^2 - br_L^2/2} = \mathcal{C}^{-1}$, which is easily solved to yield

$$r_{max} = r_L \sqrt{1 + \sqrt{1 + \frac{4\ln(\mathcal{C})k_B T}{r_L^2 \kappa_P P_0}}}. \qquad (24)$$

While the choice of $\mathcal{C}$ is arbitrary, $r_{max}$ is remarkably insensitive to this choice due to the presence of the logarithm, which is furthermore imbedded within two square roots. The utility of this approach derives from this extreme insensitivity.

With these simplifications, the upper bound to the variance of the particle position can be calculated as

$$\sigma_{U,r}^2 = \frac{1}{r_{max}} \int_0^{r_{max}} r^2 dr = \frac{1}{3} r_{max}^2 \qquad (25)$$

and that to the average power to which the particle is exposed is given by

$$P_{ave}^U = \frac{1}{r_{max}} \int_{r_L}^{r_{max}} P_0\left(\frac{r^2}{r_L^2} - 1\right) dr = P_0(r_{max} - r_L)\left(\frac{r_{max}}{3 r_L^2} + \frac{1}{3 r_L} - \frac{2}{3 r_{max}}\right). \qquad (26)$$

It is important to remember that the uniform distribution is not an exact upper limit to the exact result by virtue of the region from $r_{max}$ to $\infty$. However, it is straightforward to show that this region has a negligibly small contribution due to the rapidly decaying $\rho(r)$ for $r > r_{max}$ for sufficiently large $r_{max}$. Namely, we want to show that $\int_{r_L}^{r_{max}} e^{-ar^4 + br^2} dr \gg \int_{r_{max}}^{\infty} e^{-ar^4 + br^2} dr$. To this end, we introduce $S(r)$,

$$S(r) = \int_{r_{max}}^r e^{-ar^4 + b^2} dr. \qquad (27)$$

Using the definitions of $a$ and $b$, $S(r) = \int_{r_{max}}^r e^{-ar^2(r^2 - 2r_L^2)} dr$. For $r > \sqrt{2} r_L$, $ar^2(r^2 - 2r_L^2) > ar^2$, and $S(r) < \int_{r_{max}}^r e^{-ar^2} dr = \frac{\sqrt{\pi}}{\sqrt{a}}(\text{erf}(\sqrt{a}r) - \text{erf}(\sqrt{a}r_{max}))$. Therefore, as a limiting case, we obtain

$$\lim_{r\to\infty} S(r) < \lim_{r\to\infty} \frac{\sqrt{\pi}}{\sqrt{a}}(1 - \text{erf}(\ln\mathcal{C})) \simeq \frac{\sqrt{\pi}}{\sqrt{a}} \frac{e^{-(\ln\mathcal{C})^2}}{\mathcal{C}} \simeq 0 \text{ for } \mathcal{C} \gg 1. \qquad (28)$$

It is clear that the contribution to power exposure from the region of $r > r_{max}$, which will be neglected in the calculation of the approximate upper limit is, bounded, small in absolute terms, and significantly smaller than the overestimation that arises from the region $r_L < r < r_{max}$ for any experimentally relevant choice of the parameters.

In the case of $r_L \lesssim r_{max}$, the average power exposure is given by

$$P_{ave} = \frac{1}{r_{max}} \int_{r_L}^{r_{max}} P_0\left(\frac{r^2}{r_L^2} - 1\right) dr = \frac{P_0}{3 r_L^2 r_{max}}(r_{max}^3 - r_L^3) - \frac{P_0}{r_{max}}(r_{max} - r_L). \qquad (29)$$

Introducing $\Delta \equiv r_{max} - r_L \ll r_L$, and using the Taylor expansion,

$$r_{max}^3 = r_L^3\left(1 + \frac{\Delta}{r_L}\right)^3 \simeq r_L^3\left(1 + 3\frac{\Delta}{r_L} + 3\left(\frac{\Delta}{r_L}\right)^2 + \dots\right), \qquad (30)$$

we obtain the upper bound to the average power given by Eq. (7), i.e.,

$$P_{ave}^U = P(r_{max})\frac{\Delta}{2 r_{max}} = P(r_{max})\sqrt{\ln(\mathcal{C})\frac{k_B T}{4 r_L^2 \kappa_P P_0}}.$$

**Rate equations for the laser dynamics**. The fiber laser in our experiment consist of an Yb-doped fiber section of length $L_g$. To calculate the laser power circulation in the ring cavity of the laser, we solve the rate equations[24]. The governing equations for the pump power, $P_p$, signal power $P_s$, and amplified spontaneous emission, $P_{ASE}$, in a given longitudinal position $Z$ at the $m$th round trip are:

$$\frac{dP_{p,m}(Z)}{dZ} = \Gamma_p\left(\sigma_p^e N_{2,m}(Z) - \sigma_p^a N_{1,m}(Z)\right) P_{p,m}(Z) - \alpha_p P_{p,m}(Z), \qquad (31)$$

$$\frac{dP_{s,m}(Z)}{dZ} = \Gamma_s\left(\sigma_s^e N_{2,m}(Z) - \sigma_s^a N_{1,m}(Z)\right) P_{s,m}(Z) - \alpha_s P_{s,m}(Z), \qquad (32)$$

$$\frac{dP_{ASE,m}^{\pm}(Z)}{dZ} = \pm \Gamma_s\left(\sigma_s^{e,\pm} N_{2,m}(Z) - \sigma_s^{a,\pm} N_{1,m}(Z)\right) P_{ASE,m}^{\pm}(Z) \\ \pm 2\sigma_s^{e,\pm}\Gamma_s \frac{hc^2}{\lambda_s^3}\Delta\lambda_s N_{2,m}(Z), \qquad (33)$$

where $\sigma_s^{a(e)}$, $\sigma_p^{a(e)}$ are the absorption (emission) cross sections at signal and pump wavelengths, respectively; $\Gamma_p$ and $\Gamma_s$ stand for pump and signal filling factors; $h$ is the Plank's constant; $\lambda_s$ and $\Delta\lambda_s$ are the signal wavelength and spectral width; $N_2$ and $N_1$ are the population densities of the upper and lower lasing levels at $Z$ and are given by

$$\frac{N_{2,m}}{N_{1,m}} = \frac{R_{a,m} + W_{a,m} + W_{a,m} + A_{ASE,m}^+ + A_{ASE,m}^-}{R_a + R_{e,m} + W_{a,m} + W_{e,m} + W_{a,m} + W_{e,m} + A_{ASE,m}^+ + A_{ASE,m}^- + A_s}, \qquad (34)$$

where $A_s$ is the spontaneous transition rate, $R_{a(e),m} = P_{p,m} \frac{\Gamma_p \sigma_p^{a(e)}}{\nu_p}$ are the pump transition probabilities, $W_{a(e),m} = P_{s,m} \frac{\Gamma_s \sigma_s^{a(e)}}{\nu_s}$ are the signal transition probabilities, and $A_{ASE,m}^{\pm} = \sum_\lambda P_{ASE,m}^{\pm} \frac{\Gamma_s \sigma^{e,\pm}(\lambda)}{\lambda}$. In the ring cavity, a portion of the output power is fed back to the cavity for the next round trip and satisfy the following boundary condition:

$$P_{s,m+1}(0) = (1 - l_{scat})(1 - l_{cavity}) P_{s,m}(L_g), \qquad (35)$$

where $l_{cavity}$ and $l_{scat}$ represent optical loss of the cavity and the particle, respectively. The power in the cavity effectively reaches to a steady state, $P$, after many roundtrips, i.e., for $m \gg 1$.

**Numerical simulations**. To perform the numerical simulations, we project Eq. (8) on the Cartesian axes $x$, $y$ and $z$, to obtain a set of uncoupled overdamped Langevin equations, corresponding to the following system of coupled finite difference equations[42],

$$x_i = x_{i-1} + \gamma^{-1} F_{ot,x}(x_{i-1}, P_i)\Delta t + \sqrt{2\Delta t D} w_{x,i}, \qquad (36)$$

$$y_i = y_{i-1} + \gamma^{-1} F_{ot,y}(y_{i-1}, P_i)\Delta t + \sqrt{2\Delta t D} w_{y,i}, \qquad (37)$$

$$z_i = z_{i-1} + \gamma^{-1}\left[F_{ot,z}(z_{i-1}, P_i) - F_g\right]\Delta t + \sqrt{2\Delta t D} w_{z,i}, \qquad (38)$$

where $\mathbf{r}_i = [x_i, y_i, z_i]$ is the particle position at timestep $i$, $P_i$ is the stationary power at timestep $i$, $\mathbf{F}_{ot}(\mathbf{r}, P) = [F_{ot,x}(x, P), F_{ot,y}(y, P), F_{ot,z}(z, P)]$ is the optical force, $F_g$ is the gravity minus the buoyancy acting on the particle, and $w_{x,i}$, $w_{y,i}$, and $w_{z,i}$ are independent Gaussian random numbers with zero mean and unitary variance. The incident power on the particle, $P_i$, is updated at each timestep. The optical loss due to the trapped particle is obtained as $l_{scat,i} = 1 - P_{scat,i}/P_i$, where $P_{scat,i}$ is the power of the scattered light collected by the second collimator. This information is used together with the laser power rate equations to calculate the laser power at timestep $i$. We note that in these ray-optics simulations aberration and distortion by the glass-water interfaces is not taken into account. These effects might contribute to a weakening of the axial confinement in the experimental intracavity trap.

**3D digital video microscopy**. We have acquired videos of the particle at 50 Hz and then used standard digital video microscopy algorithms to detect its position[43]. For the measurement of the lateral position ($r$) of the particle, we have used the standard centroid algorithm (lateral resolution 20 nm). For the measurement of the axial position ($z$) of the particle, we have acquired a stack of reference images of a stuck particle as a function of its $z$-position and compare them to the image of the particle in each frame (axial resolution of 40 nm).

## Data availability

Data and resources in support of the findings of this study are available from the corresponding authors upon reasonable request.

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

## Acknowledgements

We thank Agnese Callegari, S. Masoumeh Mousavi, and Aykut Argun for useful discussions. This work is partially financed through the European Research Council (ERC) Consolidator Grant ERC-617521 Nonlinear Laser Lithography (NLL) awarded to F.Ö.I. and the ERC Starting Grant ERC-677511 ComplexSwimmers awarded to G.V.

## Author contributions

G.V., F.Ö.I., and O.M.M. had the original idea. F.K. performed the experiments with help from P.E. F.Ö.I. developed the theoretical model with help from P.E. F.K. performed the numerical simulations and analysis. G.M. helped with data and image analysis. G.V. and F.Ö.I. wrote the paper. All authors contributed with discussion and revision of the paper.

## Additional information

**Competing interests:** The authors declare no competing interests.

