## [Peer Review File · Nature Communications]

Reviewers' comments:

Reviewer #1 (Remarks to the Author):

In their manuscript "Intracavity Optical Trapping", Kalantarifard et al. present a novel optical trapping method in which they trap a dielectric particle within the laser cavity of a fibre laser. As long as the particle is close to the focus of the optical trap, the laser intensity is quite low, but due to nonlinear effects can dramatically jump if the particle begins to move from the focus. As a result, they are able to form stable optical traps at much lower powers than employed by conventional optical tweezers and with low NA optics. The authors provide a nice theoretical explanation for the operation of this device and then experimentally confirm the results of more detailed simulations. The article is well written, the method for optical trapping is truly unique and, in my opinion, the approach has much potential. Still, I have a number of questions for the authors.

Questions/Comments:

1. An important feature of a conventional optical tweezers is that it provides a linear restoring force. This is behind almost any force spectroscopy measurement. The force imposed by the fiber trap presented here is essentially zero over some range, and then becomes highly nonlinear. Could this be used to apply controlled, calibrated forces? I'm not sure how that would work. I think more of a discussion on the motivation/utility would be helpful to the reader. Magnetic tweezers, for instance, can apply controlled forces and will not photodamage a biological system. The majority of force spectroscopy measurements are probably easier performed with magnetic tweezers. Optical tweezers, however, are still the choice for fine precision measurements. But if the goal is simply to move things around without photodamage, magnetic tweezers do the job very well with a lot less effort.
2. For a Gaussian distribution, the variance or standard deviation has a clear meaning. One sigma would mean the particle spends about 68% of its time within one sigma of the distribution. At least according to your toy model, the probability density $p(r)$ is essentially a constant distribution with sharp tails, so the variance doesn't seem like the best measure here. Likewise, I'm not understanding how you get a trap strength from the variance when the potential is not at all quadratic. What am I missing?
3. pg. 13 – You remark that the variance in particle position is in good agreement with your results, then refer the reader to Fig. 3. But Fig. 3 doesn't contain the variance in position anywhere? It would be helpful to provide the simulation's prediction here as I was having trouble comparing results.
4. Ray tracing is often used to explain conventional optical tweezers as well, but the basic theoretical concepts hold for particles smaller than the trap wavelength. Is this approach relevant for small particles?
5. pg. 15 – You remark that several alternative approaches have been previously put forward to reduce laser power or to employ low NA optics. How does your approach compare? In the same paragraph you use the conjunction "but" when discussing these other approaches. This implies that there are issues with these other methods; however, I don't see what the issues are. For instance, one approach is to modulate the laser intensity so that it's turned down whenever the beam is in the center of the trap, which is similar to what you propose here, only the feedback mechanism is not built in. I think you get about an order of magnitude reduction in the necessary laser intensity, which is limited by the microsecond speeds at which you can do the feedback. Are the gains you attain simply because you essentially perform this feedback at much faster timescales (nano compared to micro)?

6. Fig. 6 I'm a bit confused about the labeling. Figure titles would be helpful. And what about the poor fit of the theory to the experiment? Why is this?

Small Errors / Typos

Pg. 3 – “There are also countless demonstrations of nonlinear interactions of lasers with materials leading to fascinating results.” This is extremely colloquial. I would suggest either removing the sentence or being more specific.

Pg. 6 – should be “increasing quadratically” not “increasingly quadratically”

Fig. 2- should be “The solid line represents the” (extra “the”)

Pg. 10 – should be “illustrates the simulations’ results” (apostrophe issue)

Pg. 11 – should be “included a fiberized band-pass filter” (“an” instead of “a”)

Pg. 11- I'm not familiar with the term “full width at half wavelength”.

Pg. 17- comma usage - “We have demonstrated through a simple analytical model, simulations of a realistic model, and experiments, a novel optical trapping scheme where the laser is...” (extra commas)

Reviewer #2 (Remarks to the Author):

The paper entitled Intracavity Optical Tweezers by F. Kalantarifard et al describes a model and experimental realisation of an optical tweezers method in which the trapping is done in a ring laser cavity. This method is based on a non-linear feedback forces using a ring cavity fibre laser and a very low numerical aperture lens. In this method the laser mode is directly influenced by the position of the particle within the cavity. The advantage of trapping performed in this way is that due to the power self-regulation due to an optomechanical coupling it is possible to reduce the average light intensity by two orders of magnitude and obtain stable trapping for particles of varying sizes – between 3 and 7 microns. This could make this method attractive for applications within biology where the high intensity of the trapping laser could potentially harm the biological entities that are being trapped. The other advantage is that the numerical aperture used here is very low so a potential application for lab on a chip and microfluidics could be done.

At first this idea looks very interesting and innovative but on reflection a number of questions can be asked: On a closer look this configuration of the trap looks very much like a commonly used counter-propagating trap. So the question first is what is the difference and also whether the authors can make a comparison between their intracavity trap and for example a mirror trap or with a more commonly used counter-propagating trap to prove whether they are not one and the same. The second thing is whether the authors can present a power spectrum to show that we are really talking here about a non-linear feedback forces. Isn't what authors do just a position clumping? The third thing is to verify the condition of a fast laser turn on and turn off. The fourth thing is that if I compare results of simulation given in figure 3 with the experimental results given in figure 5, I am not sure that I really can see such a great agreement between them as the authors believe there is. What are the error bars in figure 5? More discussion is needed on this comparison.

Some of the results of this work have been also discussed in an SPIE Optics and Photonics paper from 2013. Also further results have been presented and as an abstract for one of the OSA

conferences.

In its current form I doubt whether the paper is of the quality for publication in Nature Comm. If the authors can answer all of the questions above and make a strong argument that we really are not looking at another variation of a counter propagating beams trap then maybe it could be considered again.

Reviewer #3 (Remarks to the Author):

Kalantarifard et al use optical tweezers to trap microparticles in an aqueous solution. The particles are placed inside the optical feedback path of the trapping laser and create scattering loss dependent on their position in respect to the laser beam. The laser is operated close to the lasing threshold, which leads to a position dependent laser intensity. Compared to constant intensity tweezers, this feedback improves the particle confinement per unit intensity and enables trapping with low NA lenses.

The novelty of the paper lies in the details of the implementation of the feedback on the trapping laser intensity. Optical feedback (References 12 and 13), non-linear electronic feedback (Ref 11) and intra-cavity optical trapping (Ref 11) have been demonstrated before. Reference 12 explicitly states that the optical feedback reduces the necessary laser intensity for trapping by an order of magnitude. This paper stands out for the use of a laser gain medium inside the optical feedback path.

A discussion or experiment that compares a laser gain medium for optical feedback and electronic feedback is missing. The slow feedback bandwidth needed for the control of the particle is ideally suited for a tailored electronic feedback. The inherent speed or possibly superior noise performance of an optical feedback is not utilized. Overall, an analog or FPGA based electronic feedback schemes that measures the scattering loss or the particle's position and processes this signal in an arbitrary way seems to be more versatile and useful.

Further comments:

The title of the paper is too general. Intra-cavity trapping of atoms started in the late 90's. Since then many experiments used and reported trapping of atoms and nanoparticles in cavities.

Figure 4 compares particle confinement per unit intensity. It would be insightful to add an inset that is analogous to figure 3 c but for high-NA data. The discrepancies in Figure 4 b and d between simulation and experiment should be investigated and discussed in the text.

The authors could emphasize the importance of their work by citing specific experiments that were limited by their trapping technique and would have benefited from the techniques presented in this paper.

Reviewer 1

In their manuscript “Intracavity Optical Trapping”, Kalantarifard et al. present a novel optical trapping method in which they trap a dielectric particle within the laser cavity of a fibre laser. As long as the particle is close to the focus of the optical trap, the laser intensity is quite low, but due to nonlinear effects can dramatically jump if the particle begins to move from the focus. As a result, they are able to form stable optical traps at much lower powers than employed by conventional optical tweezers and with low NA optics. The authors provide a nice theoretical explanation for the operation of this device and then experimentally confirm the results of more detailed simulations. The article is well written, the method for optical trapping is truly unique and, in my opinion, the approach has much potential. Still, I have a number of questions for the authors.

We thank the Reviewer for recognizing the novelty and potential of the approach we propose. We answer all the questions raised by this Reviewer below.

Questions/Comments:

1. An important feature of a conventional optical tweezers is that it provides a linear restoring force. This is behind almost any force spectroscopy measurement. The force imposed by the fiber trap presented here is essentially zero over some range, and then becomes highly nonlinear. Could this be used to apply controlled, calibrated forces? I'm not sure how that would work. I think more of a discussion on the motivation/utility would be helpful to the reader. Magnetic tweezers, for instance, can apply controlled forces and will not photodamage a biological system. The majority of force spectroscopy measurements are probably easier performed with magnetic tweezers. Optical tweezers, however, are still the choice for fine precision measurements. But if the goal is simply to move things around without photodamage, magnetic tweezers do the job very well with a lot less effort.

We thank the Reviewer for asking this question. Intracavity optical trapping can be used to apply and measure microscopic forces. In fact, one can use the standard approaches to optical force spectroscopy for small displacements from the equilibrium position, where the forces associated to intracavity optical trapping are in first approximation linear. For large displacements or for a more accurate measurement of the forces, one can take into account the force nonlinearities introduced by intracavity optical trapping. Below we discuss these points in more detail.

We agree with the Reviewer that one of the most successful applications of optical tweezers has been to the measurement and application of forces. As mentioned by the Reviewer, a crucial feature that has simplified the use of optical tweezers as force transducers is the fact that the restoring force in conventional optical tweezers is linear over a certain range of displacements, depending on the particle size and the laser beam spot. Then, the use of optical tweezers as a calibrated force detector relies on the definition of an effective harmonic potential yielding a Gaussian probability distribution, a Lorentzian power spectrum density, and an exponential decay of the autocorrelation functions of the position of the trapped particle. In this context, we remark that nonlinearities (e.g., in the detector, in the optical forces for large displacements, due to the non-perfectly-spherical shape of the particle) can play a role even in a standard configuration, but can usually be safely neglected.

To explain the principle of operation of intracavity optical trapping we discuss in the manuscript a toy model that assume zero power and hence zero force, over some range of displacements, and a nonlinear increase out of this range. This simple model grasps the general features of intracavity trapping related with a much smaller power needed for its operation. However, both a full accurate modelling and the actual measurements clearly show that for *small* displacements there is a linear relation between the optical force and the particle displacement. This is observed in the analysis of the

particle tracking signals obtained from the simulations and from the experiments that we now show in the newly added Supplementary Figures 2 and 3, and in Supplementary Note 2 “Analysis of the optical potential generated by the intracavity optical trap”. We analyse both the simulated and measured particle tracking signals by obtaining their probability distribution, autocorrelation function (ACF), and power spectral density (PSD). All of these quantities show that, for small displacements, the particle displacement is consistent with an effective harmonic potential with associated meaningful transverse and axial force constants. Thus, the restoring force in intracavity trapping is linear for small displacements, while it becomes nonlinear for larger displacements from the equilibrium position.

We note that a similar situation is occurring in the case of plasmonic trapping through the self-induced back-action effect (see Juan, et al., Nat. Physics, 5, 915, 2009). In fact, despite in this case the nanocavity is passive, the relation between force and displacement is linear for small displacements and becomes nonlinear for larger ones because of the excitation of a plasmonic mode (see fig. 3e of the abovementioned article). The transition from a linear to a nonlinear regime can be observed by plotting the corresponding restoring force vs displacement (see the newly added in Supplementary Figure 4 and Supplementary Note 3 “Nonlinearity of the displacement-force curves”). In the transverse direction, a change in the slope at small displacements (below 1 micron) to larger displacements is clearly visible. Also in the axial direction, a linear range is clearly distinguished for small displacements.

In order to clarify all these considerations, we have now added the Supplementary Figures 2, 3 and 4 as well as Supplementary Note 2 “Analysis of the optical potential generated by the intracavity optical trap” and Supplementary Note 3 “Nonlinearity of the displacement-force curves”, and we have revised the text of the manuscript:

Pages 6-7: “The toy model we have discussed until now is a simplified model that describes the qualitative behavior of the intracavity optical trapping scheme as well as its nonlinear response for large particle displacements. In this model, the power and hence trapping force are considered to be zero for small particle displacements. However, in reality they have small values that do operate the trap even when the particle is near the equilibrium position. Thus, we need an accurate description of the coupling between the laser and the trapped particle thermal dynamics at equilibrium to compare with experiments. In particular, accurate simulations can help to associate an effective harmonic potential to the optical trap for small displacements from the equilibrium position, and hence to define a meaningful stiffness using the standard calibration methods based on the thermal fluctuations of a trapped particle (see Supplementary Figures 2, 3 and 4, and Supplementary Notes 2 and 3).”

Pages 8-9: “These results can be directly compared with the toy model results shown in Fig. 2. We observe that for small displacements the power is small (Fig. 2b), rather than zero, and constant (Fig. 2b, Supplementary Figure 4 and Supplementary Note 3), and the force is linear consistent with a Hookean response of the trap (Supplementary Figure 4 and Supplementary Note 3). In fact, the power is below the laser threshold in this linear regime for small displacements and increases suddenly for large displacements within the nonlinear regime. As a consequence, the corresponding probability distribution (Fig. 2d) is close to a Gaussian enabling a calibration of optical forces for small displacements (Supplementary Figure 2 and Supplementary Note 2). For large displacements nonlinear effects can be observed in the tails that yield a sub-Gaussian probability distribution (inset of Fig. 2d).”

2. For a Gaussian distribution, the variance or standard deviation has a clear meaning. One sigma would mean the particle spends about 68% of its time within one sigma of the distribution. At least according to your toy model, the probability density $p(r)$ is essentially a constant distribution with sharp tails, so the variance doesn't seem like the best measure here. Likewise, I'm not understanding how you get a trap strength from the variance when the potential is not at all quadratic. What am I missing?

As explained above, the toy model is a simplified model that describes the qualitative behavior of the intracavity optical trapping scheme and its nonlinear response for large particle displacements. However, for small displacements the toy model does not give the correct description of the system and we have therefore done a full modelling to compare with experiments. In particular, using this more detailed model, we observe that an effective harmonic potential can be associated to the optical trap, for small displacements, and, therefore, a meaningful spring constant can be defined and measured from the different calibration methods (Gaussian probability distribution, Lorentzian PSD, exponential decaying ACF, now shown in Supplementary Figures 2 and 3).

We now clarify all these considerations in the text of the revised manuscript:

Pages 6-7: “The toy model we have discussed until now is a simplified model that describes the qualitative behavior of the intracavity optical trapping scheme as well as its nonlinear response for large particle displacements. In this model, the power and hence trapping force are considered to be zero for small particle displacements. However, in reality they have small values that do operate the trap even when the particle is near the equilibrium position. Thus, we need an accurate description of the coupling between the laser and the trapped particle thermal dynamics at equilibrium to compare with experiments. In particular, accurate simulations can help to associate an effective harmonic potential to the optical trap for small displacements from the equilibrium position, and hence to define a meaningful stiffness using the standard calibration methods based on the thermal fluctuations of a trapped particle (see Supplementary Figures 2, 3 and 4, and Supplementary Notes 2 and 3).”

Pages 8-9: “These results can be directly compared with the toy model results shown in Fig. 2. We observe that for small displacements the power is small (Fig. 2b), rather than zero, and constant (Fig. 2b, Supplementary Figure 4 and Supplementary Note 3), and the force is linear consistent with a Hookean response of the trap (Supplementary Figure 4 and Supplementary Note 3). In fact, the power is below the laser threshold in this linear regime for small displacements and increases suddenly for large displacements within the nonlinear regime. As a consequence, the corresponding probability distribution (Fig. 2d) is close to a Gaussian enabling a calibration of optical forces for small displacements (Supplementary Figure 2 and Supplementary Note 2). For large displacements nonlinear effects can be observed in the tails that yield a sub-Gaussian probability distribution (inset of Fig. 2d).”

3. pg. 13 – You remark that the variance in particle position is in good agreement with your results, then refer the reader to Fig. 3. But Fig. 3 doesn't contain the variance in position anywhere? It would be helpful to provide the simulation's prediction here as I was having trouble comparing results.

We meant to say that the numerical values obtained from the experiments agree well with those obtained from the simulations in the previous section. We have now clarified this in the text of the revised manuscript, which reads:

Page 11: “[...], whose numerical values are in good agreement with the results of the simulations presented in the previous section.”

4. Ray tracing is often used to explain conventional optical tweezers as well, but the basic theoretical concepts hold for particles smaller than the trap wavelength. Is this approach relevant for small particles?

Following the Reviewer's suggestion, we now analyse the scaling of intracavity optical trapping for small particles, i.e., smaller than the wavelength. In order to understand in detail intracavity optical trapping for small particles, we exploit the dipole approximation to describe the light-particle interaction and optical forces. In particular, we shed light on the relation between extinction cross section, particle displacement, scattered power, and cavity losses as well as describe the size scaling behaviour of the relevant quantities for small particles.

We now detail the theory of optical forces in the dipole approximation and the calculations of optical losses in this regime in the newly added Supplementary Note 5 "Intracavity optical trapping for small particles". In particular, we use this approximation to calculate and plot in Supplementary Figure 6 the optical loss as a function of displacement calculated for our experimental parameters (NA=0.12, $\lambda=1030$ nm, polystyrene particles with radii of $R=300-600$ nm, Supplementary Figure 6a), and its size scaling behavior (for polystyrene particles, Supplementary Figure 6b). Since the particles are non-absorbing, the scaling rapidly decreases as R^6 for small radii. Thus, for small particles the power scattered in the trap rapidly decreases and hence the nonlinear feedback that regulates intracavity trapping is reduced. Thus, for negligible losses, i.e. $I_{\text{scat}} \approx 0$, the intracavity trap behaves as a standard single-beam optical tweezers and cannot efficiently trap small particles at low intensity. Despite the dipole approximation can be safely applied only for a particle's radius below 200 nm, we can observe how large particles are needed to increase the scattering in the trap and increase the cavity optical losses to efficiently operate the intracavity feedback trapping. This clearly shows that, for our experimental parameters, the smaller the particle the less effective the intracavity trapping. Thus, for our experimental parameters, the intracavity feedback trapping is efficient at the microscale while reduces to a standard single-beam optical trapping at the nanoscale that cannot efficiently trap nanoparticles.

However, the intracavity optical trapping approach can in principle be scaled down to particles significantly smaller than the wavelength by changing the experimental parameters. In particular, increasing the numerical aperture would allow to increase the losses due to the particle and therefore make the intracavity optical trapping more efficient at the nanoscale.

Apart from adding the new Supplementary Figure 6 and Supplementary Note 5 "Intracavity optical trapping for small particles", we now expanded the text of the main manuscript to clarify these points:

Page 9: "We note that the basic theoretical concepts hold also for particles smaller than the trap wavelength. However for our experimental parameters, the smaller the particle the less effective the intracavity trapping. In fact, it is possible to describe the light-particle interaction and optical forces in intracavity optical trapping of small particles by exploiting the dipole approximation [3, 7] (see Supplementary Note 5). Since for small particles the power scattered in the trap decreases rapidly, the nonlinear feedback that regulates intracavity trapping is reduced. Thus, for negligible optical losses, the intracavity trap behaves as a standard single-beam optical tweezers and cannot efficiently trap small particle at low intensity (see Supplementary Figure 6). For our experimental parameters, microparticles are needed to increase light scattering and hence optical losses in the trap to efficiently operate the intracavity feedback trapping. Thus, the intracavity feedback trapping is efficient at the microscale while reduces to a standard single-beam optical trapping at the nanoscale that cannot efficiently trap nanoparticles. However, the intracavity optical trapping approach can in principle be scaled down to particles significantly smaller than the wavelength by changing the experimental

parameters. In particular, increasing the numerical aperture would allow to increase the losses due to the particle and therefore make the intracavity optical trapping more efficient at the nanoscale.”

5. pg. 15 – You remark that several alternative approaches have been previously put forward to reduce laser power or to employ low NA optics. How does your approach compare? In the same paragraph you use the conjunction “but” when discussing these other approaches. This implies that there are issues with these other methods; however, I don’t see what the issues are. For instance, one approach is to modulate the laser intensity so that it’s turned down whenever the bead is in the center of the trap, which is similar to what you propose here, only the feedback mechanism is not built in. I think you get about an order of magnitude reduction in the necessary laser intensity, which is limited by the microsecond speeds at which you can do the feedback. Are the gains you attain simply because you essentially perform this feedback at much faster timescales (nano compared to micro)?

The fundamental novelty of intracavity optical trapping is that the optical forces derive from an intracavity feedback that occurs when the optically trapped particle is placed *inside* the laser cavity. As an additional benefit, this new kind of optical forces permits to achieve trapping at lower average laser intensities than alternative techniques. We discuss more in detail these differences below.

Following the Reviewer’s advice, we have significantly expanded the discussion of the comparison between intracavity optical trapping and alternative methods. In fact, there are several differences between intracavity optical trapping and alternative methods. The most important ones are:

1. Intracavity optical trapping is based on intracavity optical forces which emerge from the influence of the optically trapped particle on the laser cavity, and have a different physical origin than other kinds of optical forces. Regardless of performance, this is a fundamental difference compared to the alternative methods.
2. The feedback mechanism is build-in into the optics, so it does not require an explicit detection of the particle position. This permits the realization of an all-optical device.
3. Being the feedback due to the laser cavity dynamics it occurs on the timescale of nanoseconds, so it is several orders of magnitude faster than external feedbacks.

In particular, we compare it with self-induced back action (SIBA) trapping, counterpropagating beam trapping, and mirror optical trapping. In all these trapping techniques, a low-NA objective lens is generally used and both nanoparticles and microparticles can be confined with a relatively low intensity of laser light at the sample. Clearly each technique has its own positive and negative aspects. For example, SIBA trapping can efficiently trap nanoparticles, but is intrinsically working in close proximity with a substrate. On the other hand, counterpropagating beam and mirror trapping are specifically designed to remove radiation pressure increasing trap stability, although alignment of the beams is more critical than in other schemes.

Also, as mentioned by the Reviewer, a feedback modulation of the laser beam intensity can also reduce the intensity on average, but a high intensity is required for a time comparable to the particle Brownian dynamics (larger than milliseconds in our case) for a reasonable trap stability. We stress that intracavity optical trapping is fundamentally different from other electronic feedback-based approaches as it is an all-optical nonlinear feedback scheme that exploits a single-beam in an active laser cavity. In fact, the short time response (nanoseconds) ensures an ultra-low intensity operation that with other electronic feedback schemes is not possible.

To clarify these points, we have added a detailed discussion of the comparison between intracavity optical trapping and alternative methods in the newly added Supplementary Figure 1 and Supplementary Note 1 “Comparison of intracavity optical trapping with other optical trapping schemes”. We have also revised the text of the main manuscript:

Pages 11-12: “We remark that several alternative approaches have been proposed to make optical tweezers capable of trapping particles at low NA (see the comparison in Supplementary Figure 1 and Supplementary Note 1). Clearly, each technique has its own strengths and weaknesses with respect to intracavity trapping. For example many of these methods require the use of multiple or structured optical beams, such as in counter-propagating optical tweezers [25–28], in mirror trapping [29] and in trapping with focused Bessel beams [30], or special sample preparation, such as in trapping using self-induced back action [14, 15]. Intracavity trapping has fundamental differences in that it operates at the microscale with an all-optical nonlinear feedback scheme coupling the laser cavity with the optomechanical response of the trapped particle.”

6. *Fig. 6 I'm a bit confused about the labeling. Figure titles would be helpful. And what about the poor fit of the theory to the experiment? Why is this?*

Following the Reviewer suggestion, we have now included titles in figure 6.

Regarding the agreement between theory and experiments, we point out that (1) the theory curves have no free parameters, (2) ray optics does not provide exact predictions of optical forces, and (3) the theory does not take into account effects such as aberrations or distortions by the sample chamber that might change the trapping efficiency in certain conditions. Therefore, we consider that the overall agreement between a ray-optics theory without free parameters and experiments is quite satisfactory. In fact, for the radial direction the agreement is quite good, while the disagreement is occurring for the axial direction where aberration and gravity can play an important role in weakening the trapping stiffness. We now clarify this in the text of the revised manuscript:

Page 12: “Fig. 6 shows a comparison of the experimental and simulated confinement per unit intensity at the sample for polystyrene and silica particles of various sizes. This is a quantity that is directly related to the effective stiffness of the trapping potential explored by the particle for small displacement. The intracavity confinement per unit intensity is consistently about two orders of magnitude higher than that for standard optical tweezers. We observe that for the radial direction we find a good agreement with no free parameters between experimental and simulated results. On the other hand, for the axial direction the intracavity experimental confinement is smaller than the simulated curves. This can be accounted for by noting that simulations do not take into account effects such as aberrations or distortions by the sample chamber (glass-water interfaces) that might change the feedback and trapping point, weakening the axial trapping efficiency in experiments. Furthermore, for the axial direction ray optics might not provide a very accurate description of optical forces even for this large particle size [31, 32]. Therefore, we can consider that the overall agreement between our ray-optics theory without free parameters and experiments to be quite fair.”

Small Errors / Typos

Pg. 3 – “There are also countless demonstrations of nonlinear interactions of lasers with materials leading to fascinating results.” This is extremely colloquial. I would suggest either removing the sentence or being more specific.

We endorsed the suggestion of the Reviewer to delete this sentence.

Pg. 6 – should be “increasing quadratically” not “increasingly quadratically”

We have corrected this typo.

Fig. 2- should be “The solid line represents the” (extra “the”)

We have removed the extra “the”.

Pg. 10 – should be “illustrates the simulations’ results” (apostrophe issue)

We have corrected this typo.

Pg. 11 – should be “included a fiberized band-pass filter” (“an” instead of “a”)

We have corrected this typo.

Pg. 11- I’m not familiar with the term “full width at half wavelength”.

This is a typo, we have now corrected it to “full width at half maximum”.

Pg. 17- comma usage - “We have demonstrated through a simple analytical model, simulations of a realistic model, and experiments, a novel optical trapping scheme where the laser is...” (extra commas)

We removed the commas as suggested by the Reviewer.

Reviewer 2

The paper entitled Intracavity Optical Tweezers by F. Kalantarifard et al describes a model and experimental realisation of an optical tweezers method in which the trapping is done in a ring laser cavity. This method is based on a non-linear feedback forces using a ring cavity fibre laser and a very low numerical aperture lens. In this method the laser mode is directly influenced by the position of the particle within the cavity. The advantage of trapping performed in this way is that due to the power self-regulation due to an optomechanical coupling it is possible to reduce the average light intensity by two orders of magnitude and obtain stable trapping for particles of varying sizes – between 3 and 7 microns. This could make this method attractive for applications within biology where the high intensity of the trapping laser could potentially harm the biological entities that are being trapped. The other advantage is that the numerical aperture used here is very low so a potential application for lab on a chip and microfluidics could be done.

We thank the Reviewer for recognizing that the approach we propose is potentially attractive for biological and microfluidic applications thanks to its low power and low numerical aperture. We answer all the questions raised by this Reviewer below.

1. At first this idea looks very interesting and innovative but on reflection a number of questions can be asked: On a closer look this configuration of the trap looks very much like a commonly used counter-propagating trap. So the question first is what is the difference and also whether the authors can make a comparison between their intracavity trap and for example a mirror trap or with a more commonly used counter-propagating trap to prove whether they are not one and the same.

The fundamental difference is that in intracavity optical trapping the particle is *inside* the optical cavity of the laser and can therefore influence the laser dynamics. This is not the case with other optical trapping techniques for trapping microscopic particles, including the mirror traps and the counter-propagating traps mentioned by the Reviewer.

More specifically, the intracavity optical trap we propose is very different from the counterpropagating schemes mentioned by the Reviewer, where two beams propagate opposite to each other permitting one to reduce the scattering force pushing the particle along the beam propagation direction. In the intracavity optical trapping scheme, there is a single beam propagating in the fiber laser cavity along the direction indicated by the arrows in the revised Figures 1 and 4 and, therefore, it is essentially different from a counterpropagating trap.

To describe this comparison in more detail and more quantitatively, we added Supplementary Figure 1 and Supplementary Note 1 “Comparison of intracavity optical trapping with other optical trapping schemes” to compare our method with respect to other schemes in the literature. We also clarify this point in the in the revised text of the revised manuscript:

Pages 11-12: “We remark that several alternative approaches have been proposed to make optical tweezers capable of trapping particles at low NA (see the comparrison in Supplementary Figure 1 and Supplementary Note 1). Clearly, each technique has its own strengths and weaknesses with respect to intracavity trapping. For example many of these methods require the use of multiple or structured optical beams, such as in counter-propagating optical tweezers [25–28], in mirror trapping [29] and in trapping with focused Bessel beams [30], or special sample preparation, such as in trapping using self-induced back action [14, 15]. Intracavity trapping has fundamental differences in that it operates at the microscale with an all-optical nonlinear feedback scheme coupling the laser cavity with the optomechanical response of the trapped particle.”

2. The second thing is whether the authors can present a power spectrum to show that we are really talking here about a non-linear feedback forces. Isn't what authors do just a position clumping?

Intracavity optical trapping is fundamentally different from position clamping. In position clamping, there is an external feedback on the trap power depending on the explicit measurement of the particle position. In our scheme the feedback is intrinsic to the laser cavity, passive, extremely fast and self-regulating. We now show the power spectrum of the particle position both for the experiments and for the theory based on the ray-optics modelling in Supplementary Figures 2 and 3, and we discuss them in the Supplementary Note 2 “Analysis of the optical potential generated by the intracavity optical trap”.

We note that in our scheme the nonlinear feedback regulates the trapping dynamics occurring in the millisecond regime. However, the Brownian dynamics in the trap that shows up in the power spectral density is much slower (about 10 s) because of the low intensity regime of the laser operation at equilibrium. These different dynamics are now discussed in the newly added Supplementary Note 6 “Temporal dynamics of fiber laser, intracavity trapping, and Brownian fluctuations” and related to the new Supplementary Figures 2, 3 and 7.

We now clarify this in the revised text:

Page 10: “The response time of the laser, as measured using an acousto-optic modulator inside the cavity, is about 20 ns (see Supplementary Note 6). This is several orders of magnitude faster than the dynamics of the intracavity trapping (~100 ms, see Supplementary Figure 7) and the Brownian dynamics at equilibrium (~10 s) that shows up in the autocorrelation function and power spectrum analysis (see Supplementary Figures 2 and 3) of the trapped particle tracking signals (see Supplementary Note 2).”

3. The third thing is to verify the condition of a fast laser turn on and turn off.

The Reviewer raises a very important point, because the condition of a fast laser turn on and off is crucial in the implementation of the intracavity optical trapping. In fact, the laser dynamics are many orders of magnitude faster (nanoseconds) than the optically trapped particle dynamics (milliseconds). We now demonstrate the fast switching on and off of the laser in the Supplementary Figure 7 and in Supplementary Note 6 “Temporal dynamics of fiber laser, intracavity trapping, and Brownian fluctuations”. We performed experiment in order to measure the dynamics of the laser. To measure the response time of the laser as described in Supplementary Note 6, an AOM (acousto-optic modulator) was inserted in the cavity to create a dynamical loss instead of a trapped particle. The response of the laser to the changes (10-90%) in the given signal to the AOM was then measured using an oscilloscope. The result as expected is extremely fast as shown in Supplementary Figure 7, 20 ns. The dynamics of the laser (about 20 ns) is more than six order of magnitude faster than the dynamics of the trapping (about 100 ms) and about eight orders of magnitude faster than the Brownian fluctuation dynamics in the trap at equilibrium (about 10 s) that shows up in the autocorrelation function and PSD analysis of the tracking signals. We now clarify this in the revised text:

Page 10: “The response time of the laser, as measured using an acousto-optic modulator inside the cavity, is about 20ns (see Supplementary Note 6). This is several orders of magnitude faster than the dynamics of the intracavity trapping (~100 ms, see Supplementary Figure 7) and the Brownian dynamics at equilibrium (~10 s) that shows up in the autocorrelation function and power spectrum

analysis (see Supplementary Figures 2 and 3) of the trapped particle tracking signals (see Supplementary Note 2).”

4. The fourth thing is that if I compare results of simulation given in figure 3 with the experimental results given in figure 5, I am not sure that I really can see such a great agreement between them as the authors believe there is.

What are the error bars in figure 5? More discussion is needed on this comparison.

The qualitative and, to some degree, quantitative features of figures 3 and 5 are in good agreement. More specifically, the simulation makes some predictions about the underlying *mechanism of the intracavity trapping*, illustrated in Fig. 3, specifically:

- 1- that the power is constant and approximatively zero below a certain lateral movement of the particle and then nonlinear growth beyond that range (Fig. 3c, lower panel).
- 2- that there is power-self-regulation along the axial direction (Fig. 3c, left panel).
- 3- that r , z , and P are correlated and it can be seen from the indicated points, i, ii, iii and iv for instance (Fig. 3b). When the particle moves to the right, left and down, the power increases while by lifting the particle up, the power drops.

We observe the same qualitative behaviour in the experimental results, as shown in Fig. 5b.

We furthermore point out that (1) the simulations have no free parameters, (2) ray optics does not provide exact predictions of optical forces, and (3) the theory does not take into account effects such as aberrations or distortions by the sample chamber that might change the trapping efficiency in certain conditions. Therefore, we can consider that the overall agreement between a ray-optics theory without free parameters and experiments is quite satisfactory. In fact, for the radial direction the agreement is quite good, while the disagreement is occurring for the axial direction where aberration and gravity sag can play an important role in weakening the trapping stiffness.

We have expanded the discussion of the comparison between the numerical and experimental results, which is now described in the revised text:

Page 12: “Fig. 6 shows a comparison of the experimental and simulated confinement per unit intensity at the sample for polystyrene and silica particles of various sizes. This is a quantity that is directly related to the effective stiffness of the trapping potential explored by the particle for small displacement. The intracavity confinement per unit intensity is consistently about two orders of magnitude higher than that for standard optical tweezers. We observe that for the radial direction we find a good agreement with no free parameters between experimental and simulated results. On the other hand, for the axial direction the intracavity experimental confinement is smaller than the simulated curves. This can be accounted for by noting that simulations do not take into account effects such as aberrations or distortions by the sample chamber (glass-water interfaces) that might change the feedback and trapping point, weakening the axial trapping efficiency in experiments. Furthermore, for the axial direction ray optics might not provide a very accurate description of optical forces even for this large particle size [31, 32]. Therefore, we can consider that the overall agreement between our ray-optics theory without free parameters and experiments to be quite fair.”

5. Some of the results of this work have been also discussed in an SPIE Optics and Photonics paper from 2013. Also further results have been presented and as an abstract for one of the OSA conferences.

Both the SPIE conference proceeding in 2013 and the OSA abstract in 2012 presented some very preliminary ideas and an early attempt to realize intracavity optical trapping. In both cases, despite presenting the preliminary ideas, we did not demonstrate an intracavity optical trapping setup that was

actually working. In fact, experiments and accurate modelling proved to be much more complex and it took much effort to obtain the final conclusive results shown in our current manuscript.

In its current form I doubt whether the paper is of the quality for publication in Nature Comm. If the authors can answer all of the questions above and make a strong argument that we really are not looking at another variation of a counter propagating beams trap then maybe it could be considered again.

We hope to have fully addressed all points raised by the Reviewer and, in particular, to have highlighted how the intracavity optical trapping scheme (which, e.g., uses as *single* beam and, even more importantly, it makes use of optical forces arising because the particle is *inside* the laser cavity) is *fundamentally* different from the counterpropagating optical trapping scheme (which, e.g., requires *two* counterpropagating beams that are not influenced by the particle position).

Reviewer 3

Kalantarifard et al use optical tweezers to trap microparticles in an aqueous solution. The particles are placed inside the optical feedback path of the trapping laser and create scattering loss dependent on their position in respect to the laser beam. The laser is operated close to the lasing threshold, which leads to a position dependent laser intensity. Compared to constant intensity tweezers, this feedback improves the particle confinement per unit intensity and enables trapping with low NA lenses.

We thank the Reviewer for carefully reading the manuscript. We answer all the questions raised by this Reviewer below.

The novelty of the paper lies in the details of the implementation of the feedback on the trapping laser intensity. Optical feedback (References 12 and 13), non-linear electronic feedback (Ref 11) and intracavity optical trapping (Ref 11) have been demonstrated before. Reference 12 explicitly states that the optical feedback reduces the necessary laser intensity for trapping by an order of magnitude. This paper stands out for the use of a laser gain medium inside the optical feedback path.

We would like to highlight that the main novelty of the paper lies in the identification of a fundamentally novel mechanism for trapping microparticles using intracavity optical feedback. Therefore, this approach is fundamentally different from the techniques already demonstrated in the past. In particular, the optical feedback schemes in Refs. 12 and 13 (now Refs. 14 and 15 in the revised manuscript) are *outside* the laser cavity. By design, such schemes work well only at the nanoscale and in proximity of structured surfaces; this clearly represents an important limitation in microfluidic system or for biological samples at the microscale. Furthermore, our intracavity optical trapping scheme works with two order of magnitude less intensity than SIBA trapping. The intracavity trapping in Ref. 11 (now Ref. 13 in the revised manuscript) is a complex realization used for single atoms based on a high finesse cavity that cannot be applied at the microscale or in liquids.

We have now added a detailed comparison with other techniques in the literature is now provided in the Supplementary Note 1 “Comparison of intracavity optical trapping with other optical trapping schemes” and illustrated in Supplementary Figure 1.

A discussion or experiment that compares a laser gain medium for optical feedback and electronic feedback is missing. The slow feedback bandwidth needed for the control of the particle is ideally suited for a tailored electronic feedback. The inherent speed or possibly superior noise performance of an optical feedback is not utilized. Overall, an analog or FPGA based electronic feedback schemes that measures the scattering loss or the particle’s position and processes this signal in an arbitrary way seems to be more versatile and useful.

We would like to iterate that the central novelty of our work is the identification of a new trapping mechanism based on intracavity feedback. This is intrinsically different from any electronic feedback scheme because of the different physical origin of the feedback mechanism. In fact, there are several advantages of our scheme over any FPGA based system:

1- The power in the intracavity optical cavity is self-regulated *passively* and it is independent of the particle size, geometry, refractive index, etc. The particle itself determines how much power it needs to be trapped. Therefore, as the power on the particle reaches a certain value, the mechanism starts automatically. However, any *active* method such as using FPGA requires to be programmed precisely depending on the particle size, geometry and refractive index, for instance.

2- To modulate the power in the FPGA technique, in order to change the laser power based on the position of the particle, actively, a 3D real time particle tracking is necessary for the system to operate while the intracavity scheme is self-regulated. This significantly adds to the complexity of the implementation of such technique, and detracts from its robustness.

We now discuss this in the revised manuscript:

Page 13: “Furthermore, since intracavity optical trapping is a self-regulated mechanism, it can be designed to work for different particle types without the need to explicitly determine their properties, such as in any external feedback scheme which requires an explicit detection and identification of the particle and its properties. Thus, intracavity optical trapping is fundamentally different from other approaches as it is an all-optical nonlinear feedback scheme that exploits a single-beam in a laser cavity, and therefore opens the way to a completely new subfield within optical manipulation.”

Further comments:

The title of the paper is too general. Intra-cavity trapping of atoms started in the late 90's. Since then many experiments used and reported trapping of atoms and nanoparticles in cavities.

We endorse the reviewer’s suggestion and changed the title to be more specific:
“Intracavity Optical Trapping of Microscopic Particles in a Ring-Cavity Fiber Laser”.

Figure 4 compares particle confinement per unit intensity. It would be insightful to add an inset that is analogous to figure 3 c but for high-NA data.

We believe the Reviewer is referring to Fig. 6. We note that in the case of high-NA data we use a different setup where there is no feedback, and the power and intensity are constant. In fact, standard optical trapping with low-NA lenses is not efficient and particles cannot be trapped. Here, we wish to show that intracavity trapping can work efficiently at extremely low power, while standard optical trapping with high-NA lenses requires much more intensity to work.

We have now added Supplementary Figure 5, where we show two figures with simulations of a high-NA trap operating at a power corresponding to the intracavity trap power (about 0.12 mW) and one figure with simulations corresponding to a power of 3 mW that is close to the trapping threshold in experiments. Because the trap power is decoupled by the particle fluctuations the colour map is constant at the same value. The positional fluctuations follow the Gaussian distribution typical of an optical tweezers. Note that a real high-NA trap with 0.12 mW would not be able to trap a microparticle because the particle would escape from the central region of the trap where the restoring force is harmonic (in the simulation we assume for illustration purposes the harmonic trapping region to extend to infinity).

We now clarify this in the text of the revised manuscript:

Page 9: “Differently, for high NA optical tweezers the power is decoupled from the trapped particle fluctuations (Supplementary Note 4). High-NA optical tweezers require a minimum power that is higher than an intracavity optical trap in order to achieve stable trapping. Clearly, this power is constant as a function of the particle position (Supplementary Figure 5).”

The discrepancies in Figure 4 b and d between simulation and experiment should be investigated and discussed in the text.

As in the previous point, we believe the Reviewer is referring again to Figure 6. Regarding the agreement between theory and experiments, we point out that (1) the theory curves have no free parameters, (2) ray optics does not provide exact predictions of optical forces, and (3) the theory does not take into account effects such as aberrations or distortions by the sample chamber that might change the trapping efficiency in certain conditions. Therefore, we consider that the overall agreement between a ray-optics theory without free parameters and experiments is quite satisfactory. In fact, for the radial direction the agreement is quite good, while the disagreement is occurring for the axial direction where aberration and gravity can play an important role in weakening the trapping stiffness.

We now clarify this in the text of the revised manuscript:

Page 12: “Fig. 6 shows a comparison of the experimental and simulated confinement per unit intensity at the sample for polystyrene and silica particles of various sizes. This is a quantity that is directly related to the effective stiffness of the trapping potential explored by the particle for small displacement. The intracavity confinement per unit intensity is consistently about two orders of magnitude higher than that for standard optical tweezers. We observe that for the radial direction we find a good agreement with no free parameters between experimental and simulated results. On the other hand, for the axial direction the intracavity experimental confinement is smaller than the simulated curves. This can be accounted for by noting that simulations do not take into account effects such as aberrations or distortions by the sample chamber (glass-water interfaces) that might change the feedback and trapping point, weakening the axial trapping efficiency in experiments. Furthermore, for the axial direction ray optics might not provide a very accurate description of optical forces even for this large particle size [31, 32]. Therefore, we can consider that the overall agreement between our ray-optics theory without free parameters and experiments to be quite fair.”

The authors could emphasize the importance of their work by citing specific experiments that were limited by their trapping technique and would have benefited from the techniques presented in this paper.

The intracavity optical trapping technique enables a record low in the ratio between average laser intensity and trap stiffness, while working with low NA and a large field of view (we have added Supplementary Figure 1 and Supplementary Note 1 “Comparison of intracavity optical trapping with other optical trapping schemes” to compare this method with alternative methods available in the literature). These features can have major advantages when dealing with samples prone to photodamage, such as biological samples. In fact, biological matter is sensitive to light intensity and the typical tight focus of standard optical tweezers has detrimental effects over cell manipulation, phototoxicity, and long-term survival. Ultra-low intensity at our wavelength can grant a safe, temperature controlled environment, away from surfaces for microfluidics manipulation of biosamples. In fact, multiple mechanisms that can yield cell cycle inhibition and destruction in optical trapping. These are related with local heating of the cell [Liu et al, Biophys. J. 68, 2137-2144 (1995); Peterman et al., Biophys J. 84, 1308-1316 (2003)], induction of reactive oxygen species, light-induced protein inactivation, specific pigment absorption [Snook et al. Integr. Biol. 1, 43-52 (2009)]. An accurate study of phototoxicity on optically trapped *Saccharomyces cerevisiae* in the near-infrared has been reported by Aabo et al. [J. Biomedical Optics 15, 041505 (2010)] using a counterpropagating trap configuration, reporting that 0.7 mW (corresponding to about 0.07 mW/m²) is already sufficient to detect phototoxicity over long-term exposure time (few hours) of the cells through an accurate detection of the cell area index. The role of cell exposure time and area, and total

dose in standard optical tweezers has been also investigated in the literature [Pilat et al., Sensors 17, 2640 (2017)] where a power of 10 mW (about 22 mW/m² intensity) is reported as the minimum 3D trapping power, but a pulsed mode operation of optical tweezers has been proposed for phototoxicity reduction because of the reduced interaction area. Despite the amount of sustainable dosage by a cell or specific mechanisms involved are still debatable [Pilat et al., Sensors 17, 2640 (2017)], we verified that we can 3D-trap single yeast cells with about 0.47 mW, corresponding to an intensity of 0.036 mW/m², that is at least a tenfold less intensity than standard techniques. Thus, this is promising towards cell studies with ultra-low phototoxicity.

This discussion is now included at the end of the revised manuscript:

Pages 13-14: “One of the major advantages of the intracavity optical trapping scheme is that it can operate with very low-NA lenses, with a consequent large field-of-view, and at very low average power, resulting in about two orders of magnitude reduction in exposure to laser intensity compared to standard optical tweezers. When compared to other low-NA optical trapping schemes such as SIBA trapping [14, 15], counterpropagating beam [26–28, 33] or mirror optical trapping [29], positive and negative aspects can be considered, such as in terms of trap stiffness and average irradiance of the sample (see Supplementary Figure 1 and Supplementary Note 1). Furthermore, since intracavity optical trapping is a self-regulated mechanism, it can be designed to work for different particle types without the need to explicitly determine their properties, such as in any external feedback scheme which requires an explicit detection and identification of the particle and its properties. Thus, intracavity optical trapping is fundamentally different from other approaches as it is an all-optical nonlinear feedback scheme that exploits a single-beam in a laser cavity, and therefore opens the way to a completely new subfield within optical manipulation.

These features can yield advantages when dealing with biological samples. In fact, biological matter is sensitive to light intensity and the typical tight focus of standard optical tweezers has detrimental effects over cell manipulation, phototoxicity, and long-term survival. Ultra-low intensity at our wavelength can grant a safe, temperature controlled environment, away from surfaces for microfluidics manipulation of biosamples. There are multiple mechanisms that can yield cell cycle inhibition and destruction in optical trapping that are related with local heating [34, 35], induction of reactive oxygen species, light-induced protein inactivation, specific pigment absorption [36]. Accurate studies on *Saccharomices cerevisiae* yeast cells in near-infrared counterpropagating traps [37] and standard optical tweezers [38] have found no evidence for a lower power threshold for phototoxicity. In particular, in counterpropagating traps [37] it has been shown that 3.5 mW power (corresponding to an intensity of about 0.33 mW μm⁻²) is needed for 3D trapping of a single cell, but 0.7 mW (corresponding to about 0.07 mW μm⁻²) is already sufficient to detect phototoxicity over long-term exposure time (few hours). In standard optical tweezers [38] a power of 10 mW (about 22 mW μm⁻² intensity) has been reported as the minimum 3D trapping power, but a pulsed mode operation of optical tweezers has been observed to reduce phototoxicity because of the reduced interaction area. Despite the amount of sustainable dosage by a cell or specific mechanisms involved are still debatable [38], we observed that we can 3D trap single yeast cells with about 0.47 mW, corresponding to an intensity of 0.036 mW μm⁻², that is more than a tenfold less intensity than standard techniques.”

Reviewers' comments:

Reviewer #1 (Remarks to the Author):

I think that the authors have significantly improved the manuscript from their initial submission. Much of the difficulty I had with the first draft was that it was unclear to me what the real novelty was in their trapping scheme, besides the obvious technical innovations. However, in the revision, I feel that the authors have done a much better job of presenting the importance of the work and its potential impact on optical trapping and manipulation, which is significant. I'm happy to give my support for publishing this work.

Reviewer #2 (Remarks to the Author):

In my opinion the authors have addressed all the comments of the three referees in an in-depth way and have introduced very good changes that make this paper now of sufficient quality to be accepted for publication in Nature Communications.

Reviewer #3 (Remarks to the Author):

The optical setup uses a gain medium, therefore the optical feedback is active and not passive! Passive opto-mechanical cavity effects include self-trapping or cavity cooling.

The high optical round trip losses put their system in the bad cavity regime. The fact that the light travels around their "cavity" twice has no qualitative impact on their results. The key aspect of the presented work is the use of active optical feedback.

The advantage of the optical feedback compared to electronic feedback is its uniquely high bandwidth. However, the reported high bandwidth (~10 MHz) is not necessary for the control of slow dynamics (~10Hz). The applied optical feedback relies on the particle's scattering loss, which can also be measured with a photodiode in an electronic feedback loop. The scattering rate is important to the optical feedback system in the same way it is important to an alternative electronic feedback system.

While the paper has novelty, the application of active optical feedback to opto-mechanics of particles in liquids does not make use of its uniquely high bandwidth. The photo-damage to trapped particles can also be reduced by electronically feedbacking the trap laser intensity. Electronic feedback will provide more versatility, better performance and easier implementation. Therefore, the presented technique in this paper seems to be not particularly useful.

I would recommend publication in a more specialized optics journal. I do not recommend publication in nature communications.

Manuscript NCOMMS-18-31077-A

Intracavity Optical Trapping of Microscopic Particles in a Ring-Cavity Fiber Laser

F. Kalantarifard, P. Elahi, G. Makey, O. M. Maragò, F. O. Ilday, Giovanni Volpe

Reviewer 1

I think that the authors have significantly improved the manuscript from their initial submission. Much of the difficulty I had with the first draft was that it was unclear to me what the real novelty was in their trapping scheme, besides the obvious technical innovations. However, in the revision, I feel that the authors have done a much better job of presenting the importance of the work and its potential impact on optical trapping and manipulation, which is significant. I'm happy to give my support for publishing this work.

We thank the Reviewer for their positive assessment and support.

Reviewer 2

In my opinion the authors have addressed all the comments of the three referees in an in-depth way and have introduced very good changes that make this paper now of sufficient quality to be accepted for publication in Nature Communications.

We thank the Reviewer for the positive assessment and support.

Reviewer 3

The optical setup uses a gain medium, therefore the optical feedback is active and not passive! Passive opto-mechanical cavity effects include self-trapping or cavity cooling. The high optical round trip losses put their system in the bad cavity regime. The fact that the light travels around their "cavity" twice has no qualitative impact on their results. The key aspect of the presented work is the use of active optical feedback.

We agree with the reviewer: The optical feedback is active and the fact that this produces intracavity optical forces is the novelty of the article. In fact, we state already in the introduction (page 3) that "We achieve intracavity optical trapping inside an active laser cavity [...]". Also, we would like to remark that we never used the word "passive" in the article and we only used the word "active" in the instance mentioned above. We used the words "active" and "passive" in our first Resubmission Letter referring to "extrinsic" or "intrinsic" feedback; we apologize for the confusion.

The Reviewer statement "*The high optical round trip losses put their system in the bad cavity regime*" is not generally correct. As shown in the Supplementary Figures 4c and d, when the particle moves to the centre of trap, the laser drops significantly, but if the particle moves away, the laser power is strong. In other words, the cavity finesse is strongly conditioned to the particle's position, which is the essence of the novelty of this work.

The advantage of the optical feedback compared to electronic feedback is its uniquely high bandwidth. However, the reported high bandwidth (~10 MHz) is not necessary for the control of slow dynamics (~10Hz). The applied optical feedback relies on the particle's scattering loss, which can also be measured with a photodiode in an electronic feedback loop. The scattering rate is important to the optical feedback system in the same way it is important to an alternative electronic feedback system.

We would like to point out that the main premise of our article is the conceptually novel technique of intracavity optical trapping and the introduction of *intrinsic* nonlinear feedback forces, i.e., a force that arises only because the optically trapped particle is placed *inside* a laser cavity. Thus, the intracavity optical trapping is fundamentally different from electronic feedback-based approaches.

The novelty of this approach opens many new possibilities to be explored in future works. One of these is to study systems where the characteristic timescales associated to the laser and to the motion of the optically trapped particle become comparable. This could be relevant, e.g., when studying optical trapping in air or in vacuum. However, this is beyond the scope of the present work. We now remark this in the revised manuscript (page 14):
“It is also tantalizing to consider the interplay that might arise when the timescales of the laser and of the particle dynamics become comparable, e.g., when trapping particles in air or in vacuum.”

While the paper has novelty, the application of active optical feedback to opto-mechanics of particles in liquids does not make use of its uniquely high bandwidth.

We are glad that the Reviewer acknowledges the novelty of our work and its “*uniquely high bandwidth*”, but appears to be concerned that electronic feedback could provide similar advantages. In this regard, we can only iterate that the present manuscript is the first introduction of a new trapping technique relying on intracavity feedback. As all new techniques, it will find multiple applications that take advantage of its unique features. In particular, we expect “*its uniquely high bandwidth*” to be crucial when dealing with particles to be trapped in air or in vacuum, as we now remark in the revised manuscript (page 14):
“It is also tantalizing to consider the interplay that might arise when the timescales of the laser and of the particle dynamics become comparable, e.g., when trapping particles in air or in vacuum.”

The photo-damage to trapped particles can also be reduced by electronically feedbacking the trap laser intensity.

We agree that electronic feedback can also potentially reduce photodamage, but we also believe this is not the central point of our work. On a fundamental level, the novelty of our work is the identification of the mechanism for trapping microparticles using intracavity optical feedback. This being a new mechanism, we believe it can lead scientists and engineer

to new and better applications beyond what can be achieved and possibly imagined with electronic feedback (or other established techniques, for that matter).

Electronic feedback will provide more versatility, better performance and easier implementation. Therefore, the presented technique in this paper seems to be not particularly useful.

Our manuscript reports on a new concept that, in its first ever experimental implementation, should not, in fairness, be judged purely in terms of its technical advantages/disadvantages against established techniques, which have enjoyed years of cumulative efforts by many research groups to reach their present state of technical maturity. However, even if we do adopt such a purely bottom-line perspective, we would like to respectfully disagree on all three accounts:

Regarding *versatility*, our technique is all optical, whereas electronic feedback requires separate and often dedicated electronics. Also, in our case the laser's response to the particle position is intrinsic to the laser cavity dynamics and it does not require recalibration each time experimental details are modified.

In terms of *performance*, as we have shown experimentally, our technique achieves excellent results in capturing and trapping particles, while achieving a *major reduction in optical power exposure compared to all alternative techniques*. In contrast, successful external electronic feedback would require extremely careful elimination of potential systematic errors and deleterious effects of in-loop noise. The standard benchmark for any electronic feedback system is to measure both in-loop and out-of-loop noise to ascertain its performance, adding a further layer of complication. All of these steps are non-trivial and introduce significant complications, which should be contrasted to the simplicity and, if the reviewer would allow it, the elegance of our technique.

Our technique is much *easier to implement*. The feedback mechanism is built in. At this point, we would like to kindly remind the reviewer that while fast electronics is relatively easy to attain, *low latencies*, as would be required, are much harder to achieve reliably, not to mention that in our method, explicit detection of the particle position (and its repeated calibration) is not required, unlike electronic feedback.

A brief version of this discussion has been included in the revised manuscript on pages 13-14:

“Another advantage of the intracavity optical trapping scheme is that it intrinsically features a high bandwidth thanks to the intracavity feedback. [...] In summary, intracavity optical trapping is all-optical and easy to implement as the feedback mechanism is intrinsically built in; furthermore, it has no need for separate electronics or recalibration.”

I would recommend publication in a more specialized optics journal. I do not recommend publication in nature communications.

We believe that dissemination of this work to the broad interdisciplinary audience of Nature Communication is warranted because we report on a fundamentally different mechanism for trapping microparticles using intracavity optical feedback, which, also given the ever-growing interest in optical manipulation, can find applications in a broad range of fields.